# Honey bee food resources under threat from climate change

Andreia Quaresma [1,2,3,4], Johannes M. Baveco[5], Robert Brodschneider[6], Willem Bastiaan Buddendorf [5], Norman L. Carreck [7], Kristina Gratzer [6], Fani Hatjina [8], Ole Kilpinen[9], Ivo Roessink [6], Flemming Vejsnaes[9], Jozef van der Steen[10], M. Alice Pinto [1,12] ✉ & Alexander Keller [11,12] ✉

Plant-pollinator interactions are essential for plant productivity but face growing threats from climate change, including vegetation loss and mismatches in flowering. Yet, the consequences for bee food resources remain poorly understood at continental scales. Here, we analyse 2 500 samples collected by honey bees (*Apis mellifera*) between May and August 2023 from 310 locations across Europe using ITS2 metabarcoding. We derive climatic response curves of floral resources and assess exceedance risks of interaction loss under projected climate scenarios. Our findings reveal that rising temperatures and reduced precipitation decrease the diversity of foraging resources across Europe, pushing many plants beyond critical limits. When both warming and drying coincide, the potential for resilience through temporal or spatial buffering is strongly constrained. These declines pose serious risks to bee nutrition, ecosystem functioning, and food security. Our study underscores the urgency of mitigating climate change to preserve vital plant-pollinator systems and the services they sustain.

Plant-pollinator relationships are fundamental biotic interactions in ecology due to their key role in plant productivity, food webs, and human well-being[1,2]. Animal pollinators, principally insects, contribute an estimated $235–$577 billion annually to the global economy by providing pollination services to an estimated 87.5% of the world's flowering plants and 75% of major global food crops[1,3]. However, without suitable and abundant plants supplying pollen and nectar, many pollinator populations will decline along with their vital services. An increase in temperature has already been reported to adversely impact vegetation composition[4,5] and is known to cause flowering and spatial shifts, which may lead to interaction mismatches between plants and pollinators[6–8]. The decrease in floral diversity is a key factor driving bee diversity loss[9,10], and evidence correlates bee pollinators and wild plants as being tightly linked and in parallel decline[11,12].

Several studies project that climate change will affect the survival of many plant species[13–17]. Depending on the climate change scenario, 3-21% of all endemic plant species in Europe could face extinction by 2050[13]. Under the most severe scenario, 22% of total plant species would become critically endangered and 2% extinct by 2080[14]. However, the impact of climate change varies by region, with plant diversity projected to increase in Northern Europe while declining in the Mediterranean region[15]. Furthermore, by the end of the 21st century, many

[1]CIMO, LA SusTEC, Instituto Politécnico de Bragança, Campus de Santa Apolónia, Bragança, Portugal. [2]Departamento de Biologia, Faculdade de Ciências da Universidade do Porto, Porto, Portugal. [3]CIBIO, Centro de Investigação em Biodiversidade e Recursos Genéticos, InBIO Laboratório Associado, Campus de Vairão, Universidade do Porto, Vila do Conde, Portugal. [4]BIOPOLIS Program in Genomics, Biodiversity and Land Planning, CIBIO, Campus de Vairão, Vila do Conde, Portugal. [5]Wageningen Environmental Research, Wageningen, the Netherlands. [6]Department of Biology, University of Graz, Universitätsplatz 2, Graz, Austria. [7]Carreck Consultancy Ltd, Shipley, West Sussex, UK. [8]Department of Apiculture, Institute of Animal Science ELGO 'DIMITRA', Nea Moudania, Greece. [9]Danish Beekeepers Association (DBF), Sorø, Denmark. [10]Alveus AB Consultancy, Oisterwijk, Netherlands. [11]Cellular and Organismic Networks, Faculty of Biology, Ludwig-Maximilians-Universität München, Planegg-Martinsried, Germany. [12]These authors jointly supervised this work: M. Alice Pinto, Alexander Keller. ✉e-mail: apinto@ipb.pt; keller@bio.lmu.de

plant species are expected to disappear from over half of their current geographic ranges in Europe[16,17], with distribution shifts northwards and uphill[16–18] or suffer local or global extinction[13]. These changes will severely affect the availability of food resources to pollinators. A decline in floral diversity may limit essential nutrients for pollinators, affecting their overall health and reproductive success[19]. Reduced access to diverse pollen sources can lead to nutritional deficiencies, weakened immune systems, and increased vulnerability to pathogens and parasites, as well as other environmental stressors[2]. Additionally, shifts in floral availability may alter foraging behaviour, forcing pollinators to expand their foraging ranges[20]. These disruptions may ultimately contribute to population declines, reduce pollination efficiency, and negatively impact both wild plant communities and agricultural yields for pollinator-dependent crops[2,19].

Our study investigates the impacts of climate change on the food resources of the honey bee *Apis mellifera*, the dominant global pollinator with the broadest geographical and temporal range in Europe. Although *A. mellifera* is not in decline[21], we select it due to its ubiquity, generalised foraging, and critical role as a pollinator in agroecosystems. This species' extensive spatial and temporal activity makes it a useful system for evaluating large-scale patterns in floral availability under climate change, especially in managed landscapes. The major goal of our study was to quantify the expected impact of climate change on honey bee foraging across different regions of Europe and assess the extent to which resilience may be achieved through spatial or temporal shifts in floral availability. To achieve this, we conduct a systematic survey of honey bee-collected pollen at unprecedented spatial and temporal scales. We analyse 2 500 pollen samples collected by honey bees biweekly, for nine sampling rounds, between May and August 2023, from 310 apiaries distributed across all 27 EU countries (Fig. 1A). We identify the pollen composition of these samples using DNA metabarcoding of the internal transcribed spacer 2 (ITS2)[22]. We structure our analysis around five core objectives: (1) assess spatial and temporal patterns of pollen diversity, composition and crop contribution across Europe in relation to climatic gradients, to establish a baseline for evaluating potential impacts of climate change on floral resource availability; (2) quantify historical variation in temperature and precipitation at the local study sites to contextualize and scale future projections; (3) determine the realised climatic niches of foraged pollen plant species by estimating empirical climatic niche response curves to assess their sensitivity to climate change; (4) quantify potential floral resource loss under different climate projection scenarios; and (5) evaluate the resistance and resilience of plant-pollinator interactions by spatial or seasonal compensation through alternative floral resources. Together, these objectives provide an integrated framework for understanding how climate-driven environmental change may affect floral resource dynamics and, consequently, honey bee foraging potential across Europe.

## Results and Discussion
### Pollen diversity, composition, and crop contribution
As a baseline for understanding how floral resource availability may shift under future climate scenarios, we first assessed how honey bee-collected pollen diversity and composition varied across Europe in relation to key climatic gradients. We found that temperature and precipitation gradients, as well as latitude, had significant effects on pollen Effective Species Richness ($e^H$), with a mid-latitude peak particularly evident in late summer. However, these effects had limited explanatory power (quadratic linear model with latitude and month as predictors; $e^H$ log-transformed: $F_{(3, 2493)} = 90.27$, $p < 0.001$, adjusted $R^2 = 0.10$; Fig. 1A, B, Supplementary Fig. 1). Seasonal models confirmed that this mid-latitude peak was most pronounced in July and August.

The proportion of pollen from crops per sample was generally low, with an overall average of 19% (20% standard deviation, Fig. 1C), but with peaks during mass-flowering events when crops contributed up to a mean of 50%. These peaks were primarily driven by taxa such as

*Brassica spp.*, *Helianthus annuus*, and *Zea mays* identified among the 25 most abundant pollen sources (Supplementary Fig. 2). Even during such episodic crop dependencies, honey bees maintained a comparably diverse diet at all sites throughout the season, underscoring the importance of wild floral resources (Fig. 1A)[19,20].

Pollen composition varied markedly across Europe and over the sampling season, reflecting shifts in the identity of plant species foraged upon (visualised via NMDS; Supplementary Figs. 3 and 4). To quantify how this compositional turnover related to environmental gradients, we used Generalised Dissimilarity Modelling (GDM), which explained 48.47% of the total variance in pollen composition (pseudo-$F_{(3, 48\ 205)} = 0.52$, $p < 0.001$; Supplementary Fig. 1), with contributions of 27.53% from temperature (Σ I-spline = 1.82), 14.14% from geography (Σ I-spline = 0.94), and 6.79% from precipitation (Σ I-spline = 0.44). These results indicate that floral resource composition is strongly structured by climatic and spatial factors.

### Impact of regional warming
To estimate the potential vulnerability of current floral resources under future warming, we next examined regional temperature trends and their effects on the climatic suitability of plant species foraged by honey bees at observed sites. Warming and changes in precipitation can lead to several impacts on bee resources: firstly, they can alter the local plant composition; secondly, they can disrupt plant phenology, affecting key events such as flowering timing; and thirdly, they can impact the quantity and quality of resources[6,23]. These effects can lead to mismatches in plant-pollinator interactions, changing pollen's temporal and spatial availability. We were thus interested in how future scenarios might pose risks for the currently available pollen resource under climate change and continuing historical trends. Global warming is projected to reach 2 °C by 2040 to 2060, even under aggressive mitigation scenarios[24]. However, such global trajectories can mask substantial regional variation, which is critical for forecasting. We therefore included an analysis of historical warming trends at our actual sampling sites (Supplementary Fig. 5A and 5B) to ground our projections in empirically observed, region-specific climate change and contextualise the effect size of projections. Based on the results of these analyses, sites in Southern Europe have, on average, warmed by 1.8 °C over the past 50 years, Central Europe by 1.6 °C, and Northern Europe by 0.2 °C (Supplementary Fig. 5A and 5B). A previous study with regional forecasts predicts a 2 °C increase in Northern Europe by 2040 and Central and Southern Europe by 2030, ahead of global averages[25].

To study the impacts of rising temperatures and decreasing precipitation on bee pollen resources, we first estimated temperature and precipitation response curves for each plant taxon in the dataset based on their sampling time, location, and corresponding climatic values (for more details, see Methods). We then incrementally adjusted the mean temperature (0 to +5 °C) and precipitation (0 to −50 mm) for each location and time. Using these updated climatic conditions, we assessed whether conditions exceeded the percentile thresholds of each taxon's empirical response curve and thereby whether the conditions remained suitable or became unsuitable. Our results revealed a severe impact of temperature increases on current bee food resources (both wild plants and crops) across Europe, with a gradual reduction in impact from Southern to Northern Europe (Fig. 2; Linear model $F_{(5, 1189)} = 100.3$, $p < 0.001$, adjusted $R^2 = 0.30$; Supplementary Table 1). Southern European wild plants are most affected, with Mediterranean islands such as Cyprus, Crete, and Malta facing significant risks even with just 1 °C of warming, in which about half of the current bee resources become at risk (Fig. 2, Supplementary Fig. 6A). This trend worsens for higher temperatures. In comparison, Northern Europe is less affected up to a 2 °C rise, although with a notable increase in risk under more severe warming scenarios.

These findings align with Bakkenes et al.[15], who noted that Southern European plant species are more sensitive to warming than

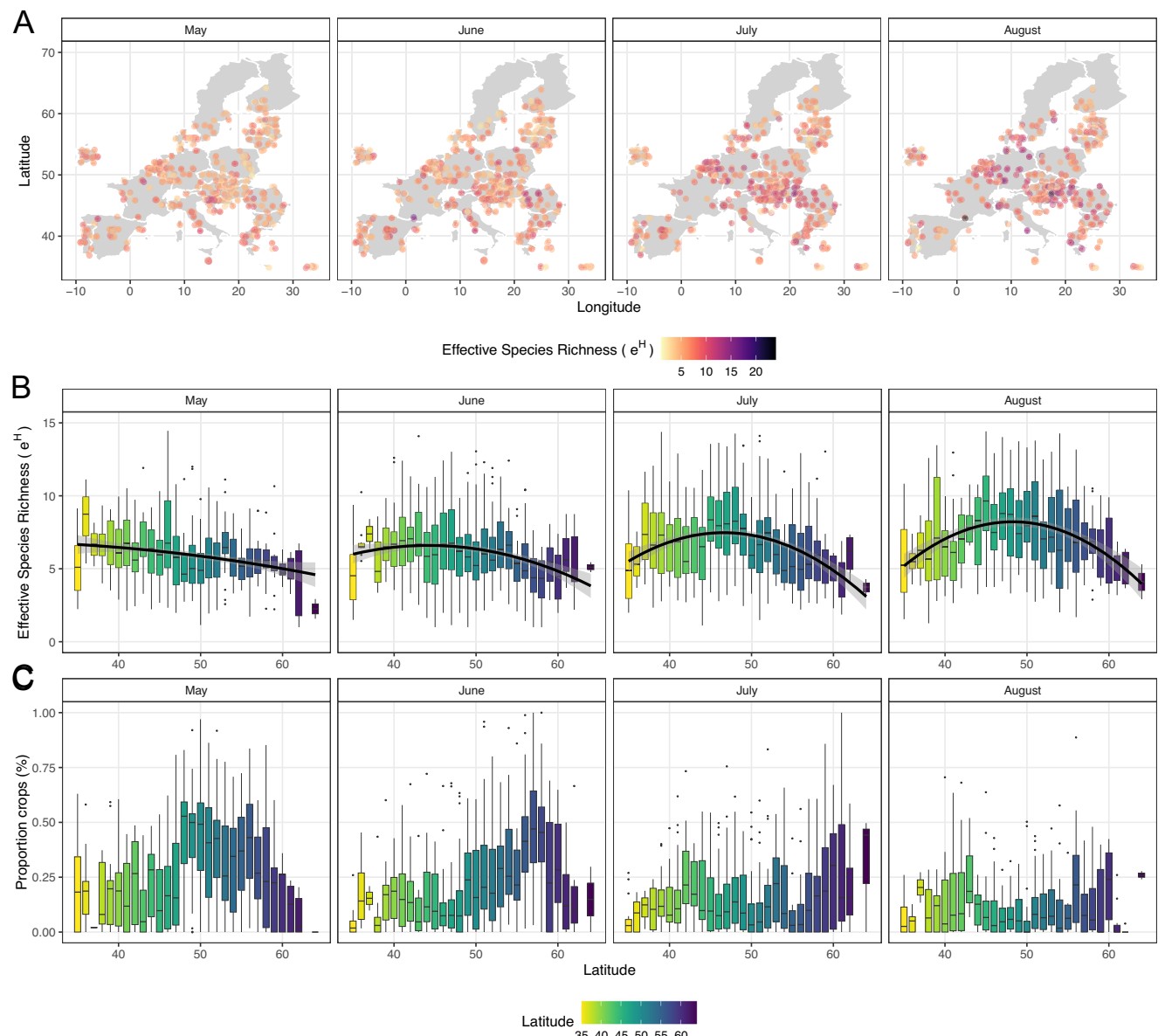

**Fig. 1 | Honey bee collected pollen diversity across different latitudes and sampling months. A** The map displays the location of sites used for collecting pollen samples collected across the 27 EU countries, including Cyprus and Malta, as well as the French island of Corsica and the Greek islands of Chios, Crete, Lesvos, and Samos, with colour representing observed Effective Species Richness ($e^H$). The underlying map was created using the *map_data()* function in the R[54] package ggplot2[62]. **B** Boxplots illustrate pollen $e^H$ variation by month and latitude. Central Europe shows higher diversity overall (two-sided multifactorial linear model with latitude as quadratic term: $F(3, 2493) = 90.27$, $p < 0.001$, adjusted $R^2 = 0.10$), with peak diversity in late summer months (July: adjusted $R^2 = 0.10$, $F(2, 685) = 37.41$, $p < 0.001$; August: adjusted $R^2 = 0.11$, $F(2, 512) = 31.40$, $p < 0.001$) compared to early summer (June: adjusted $R^2 = 0.04$, $F(2, 697) = 16.51$, $p < 0.001$) or spring (May: adjusted $R^2 = 0.04$, $F(2, 591) = 12.06$, $p < 0.001$). Error ribbons correspond to the

standard error (SE) of the fitted linear model. **C** The proportion of crop pollen across latitudes and months is shown, with crop pollen contributing less than 25% in most regions. However, during peak crop flowering seasons, particularly in Central Europe in May, crop pollen can reach up to 50%. This contribution shifts to Northern latitudes in June and July, but wild plants remain a key resource for honey bees. In (**B**) and (**C**), boxplots are based on $n = 2500$ (May: $n = 595$, June: $n = 702$, July: $n = 688$, August: $n = 515$) independent samples (biological replicates; individual honey bee foraging samples collected across sites and months), display the median (centre line), the 25th and 75th percentiles (bounds of the box), and whiskers extending to the smallest and largest values within 1.5 x the interquartile range (IQR). Outliers beyond this range are shown as individual points, and colours indicate latitude. *P*-values are exact unless smaller than 0.001, in which case they are reported as $p < 0.001$.

Northern ones. Seasonal variations revealed, in general, a higher risk during summer months compared to spring (Fig. 3A, Supplementary Fig. 7A). However, this effect did not persist at 5 °C warming, when a high proportion of plants became at risk throughout the season. Temperatures warmer than their critical limits negatively influence the growth and reproduction of species, contributing to biodiversity loss. Projections suggest an increased risk for European species[7,17,26,27] and the possible extinction of 3-21% of endemic plants by 2050[13]. Our empirical data on temperature niches support this alarming notion of

the sensitivity of bee pollen resources, with even higher proportions at risk on local scales.

## Impact of changing regional precipitation patterns
In parallel, we assessed how changes in precipitation, both observed and projected, might alter the climatic suitability of pollen-providing plants at observed sites, again focusing on species-level risk under plausible climate futures. Historically, precipitation patterns have also changed, and it is projected that precipitation will severely change

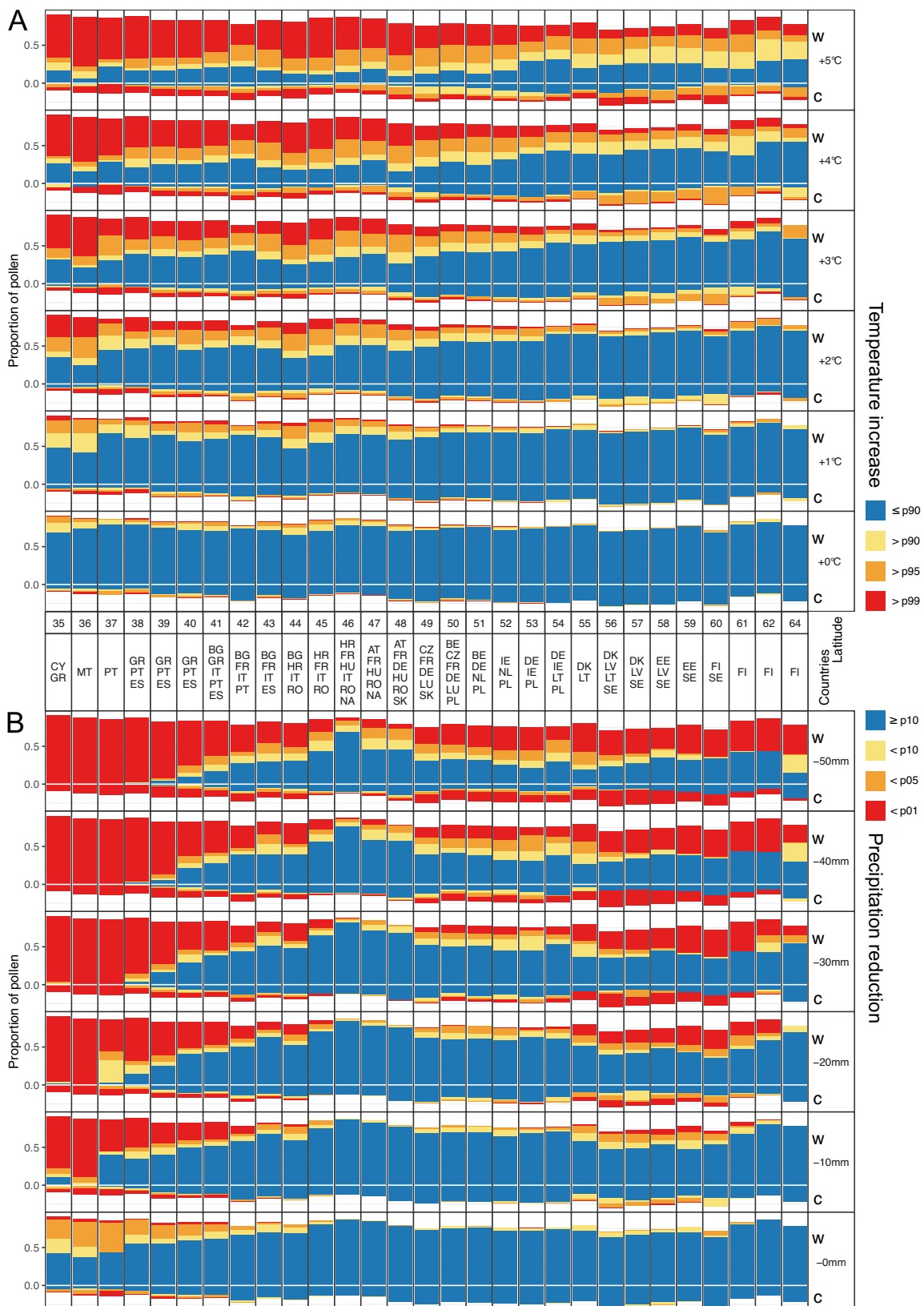

across most of Europe with climate change[28,29] (Supplementary Fig. 5C and 5D). For example, Vautard et al.[28] predict a 10% reduction in annual mean precipitation for Southern Europe and an over 10% increase for Northern Europe for the 2 °C warming scenario. However, the effects of precipitation on bee pollen resources are more complex than those of temperature, and annual mean values are not a good proxy for local water availability for plants: the increased intensity and frequency of extreme precipitation events influence annual means to a large degree, but the water will not necessarily be available for plants over extended periods, and direct harm can be caused by flooding[24]. Therefore, we focused on the impacts of a gradual decrease in mean precipitation at a local scale, excluding short-term extreme rainfall

**Fig. 2 | Impact of temperature rise and precipitation reduction on climate-related risks to honey bee pollen resources across latitudes. A** shows projected effects of a up to +5 °C temperature rise, while **B** depicts the impact of a 50 mm reduction in precipitation. **A** and **B** both highlight changes in the risk classification of honey bee pollen resources under these climate change scenarios, with a focus on different latitudinal regions. Each bar shows the cumulative relative abundance of all samples from that latitude under the given scenario, partitioned into risk classes. Crop (c) taxa are plotted as negative values and wild (w) taxa as positive values around a zero baseline (white line). Risk classification was based on kernel density estimates of each plant taxon's climate response curve, calculated from metabarcoding-based pollen abundance data matched to sampling time and local climatic conditions. Percentile-based thresholds were used to quantify whether projected future conditions would exceed species-specific climatic niches. Bar colours indicate risk levels: blue for low risk, yellow for moderate, orange for high, and red for critical risk. Country abbreviations follow ISO two-letter codes. The figure displays a subset of climate scenarios for interpretability; additional scenarios are provided in Supplementary Fig. 6.

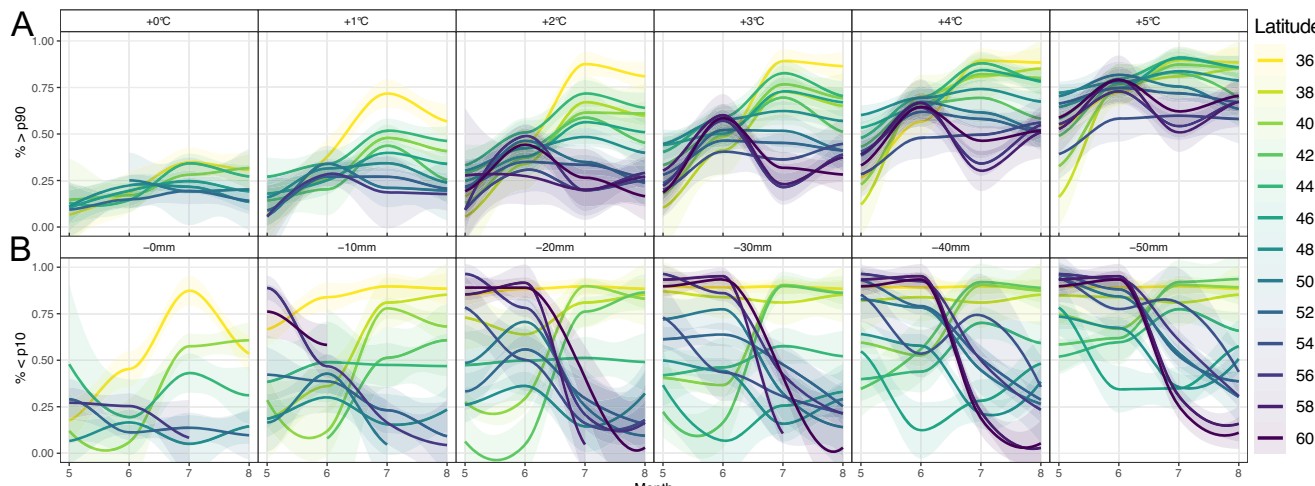

**Fig. 3 | Impact of temperature increase and precipitation decrease on the proportional risk to honey bee pollen resources across different latitudes and sampling months. A** depicts the effect of a temperature rise, and **B** shows the impact of a decrease in precipitation on the proportion of pollen resources at risk. Metabarcoding data were used to estimate the temperature and precipitation kernel-based response curves for each plant species in the dataset. This involved mapping plant taxon abundance with sampling time and location, as well as corresponding climatic conditions. Percentile exceedance analyses were conducted on these niche response curves to assess whether future climatic conditions would remain suitable for each plant taxon: greater than the 90th percentile threshold for temperature and less than the 10th percentile threshold for precipitation. These thresholds represent conditions likely unsuitable for the species under projected climate change scenarios. Each line in the graph represents the trend in pollen resources classified as "at risk" over time for different latitudes. The higher the line, the greater the proportion of pollen resources that are potentially at risk due to the projected temperature or precipitation changes. Error bands with lighter background shades represent 95% confidence intervals. For some latitude × month × scenario combinations, trend lines are absent because no pollen resources were classified as at risk under those conditions. Each scenario aggregates $n = 22\,809$ independent values. More detailed scenarios with finer-scaled increments in temperature and precipitation changes are shown in the Supplementary Fig. 7.

events. Based on species-specific climatic response curves, reduced precipitation markedly increased risks for local plants across all latitudes, with the most severe impacts in Southern Europe but also notable effects in Northern and Baltic regions (Fig. 2; linear model $F(5, 1138) = 56.9$, $p < 0.001$, adjusted $R^2 = 0.20$; Supplementary Table 1). Northern countries were particularly sensitive to spring rainfall losses, while Southern countries were more affected by summer rainfall deficits (Fig. 3B, Supplementary Fig 7B). This pattern aligns with the Copernicus Climate Change Service's 2023 report[16], which predicts increased summer precipitation in Northern Europe and decreased precipitation in Central and Southern Europe for the 2 °C warming scenario. Our results also indicate that Southern Europe, especially the Algarve region (latitude 37°) in Portugal, is already experiencing drought stress with a visible impact on pollen resources, whereas the impact at latitude 46° was less pronounced compared to the rest of Europe (Fig. 2, Supplementary Fig. 6B).

## Resistance and resilience
Climate change impacts on a local scale can, however, alter plant species distributions and phenology in various ways, so a high risk for local populations may not necessarily reduce overall plant resources[7,17,30]. Affected species may, for example, be replaced by other taxa that can provide suitable resources for honey bees by immigration or changes in flowering phenology. We thus also estimated the proportion of at-risk plants for precipitation and temperature in a combined analysis that could potentially be replaced by species better adapted to the new climatic conditions available in the wider region and/or other seasons. This assessment of resistance (non-risked species) and resilience (availability of replacement species) indicated that small changes in temperature and precipitation might be mitigated by the broader regional species pool (Fig. 4, Supplementary Fig. 8), but the combined effects of temperature increase and precipitation reduction limit this potential. Severe changes, particularly with this combination, create conditions that show low resistance and resilience, i.e. local populations are highly affected and are difficult to fill with replacement taxa. This is likely to lead to significant losses in plant diversity and honey bee food resources (Fig. 4). Our results are limited to the pollen collected, which may not represent the full resources spectrum at the study sites. Consequently, our findings might underestimate overall resilience. Conversely, unpreferred plants included in the analyses contribute equally to resilience calculations as preferred ones, despite potentially lower nutritional quality or abundance. This could lead to an overestimation of resilience. Therefore, uncertainties in both directions must be considered when interpreting our results. Despite these limitations, we can confidently state that climate change will disrupt local pollination networks particularly when temperature increases are coupled with precipitation losses, with severe implications for plants, honey bee nutrition, and competition with wild pollinators. The limited buffering capacity under combined stress conditions suggests that for many

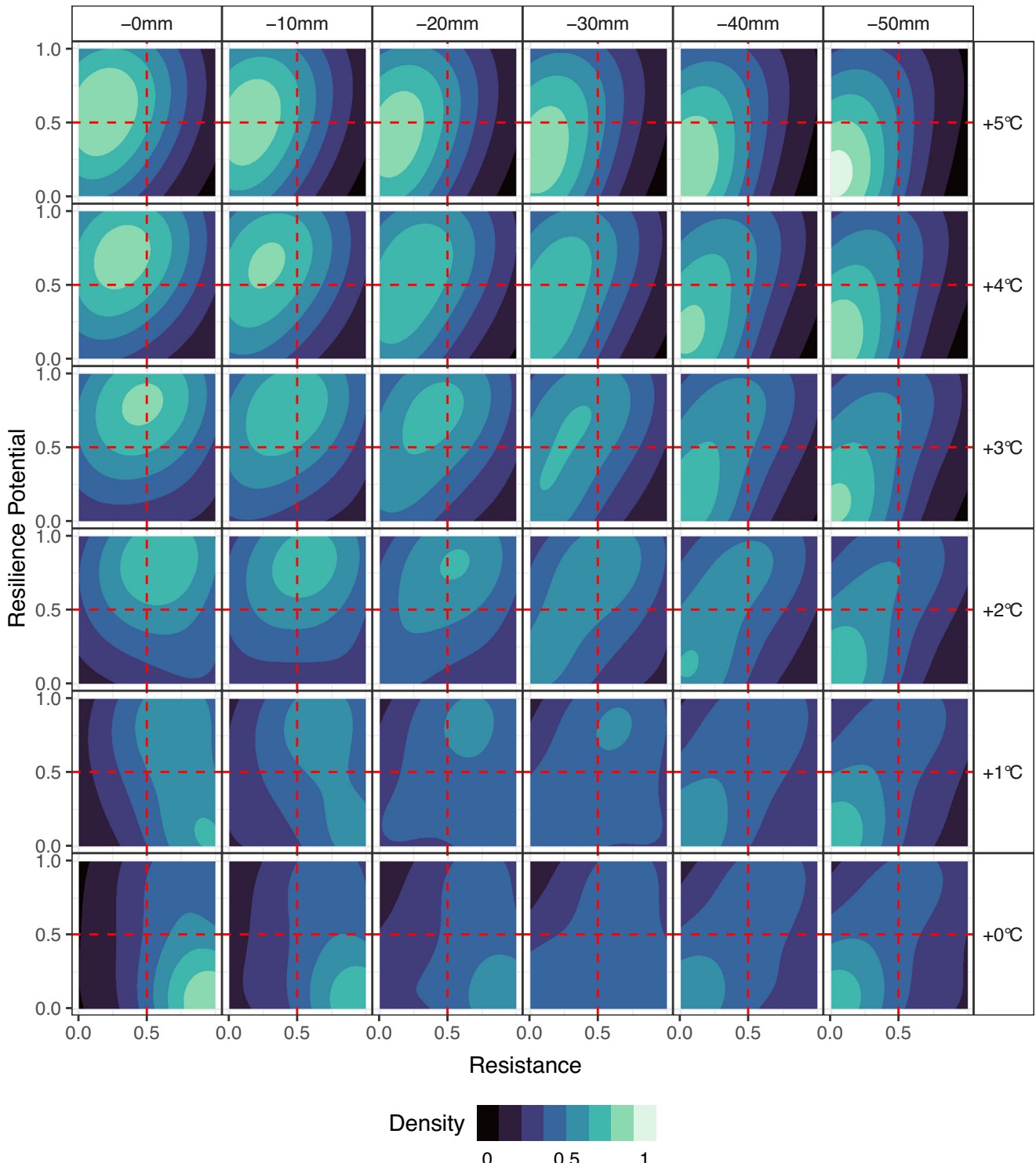

Density

0          0.5          1

**Fig. 4 | Resistance (x) and resilience (y) of honey bee food resources under various climate change scenarios combining increased temperature and decreased precipitation.** The plot shows the density distribution of sample resistance (the proportion of taxa not at risk at a given location and time) and sample resilience (the ability to compensate for losses in taxa at risk with other plant taxa). Each density plot aggregates $n = 2\,500$ independent values. The lower right quadrant highlights high resistance, with many taxa unaffected. The upper right quadrant represents high resistance and high resilience, where many taxa remain unaffected, and losses are effectively compensated. The upper left quadrant reflects low resistance but high resilience, suggesting significant plant turnover that mitigates resource scarcity. The lower left quadrant indicates low resistance and low resilience, signalling severe stress on resources available for honey bees. Density is represented by colour, with lighter colours indicating higher sample density in the distribution. The various climate change scenarios highlight the profound impact of rising temperatures combined with decreasing precipitation on the ecosystem's ability to sustain a diverse array of honey bee plant resources. More detailed scenarios with finer-scaled increments in temperature and precipitation changes are shown in the Supplementary Fig. 8.

sites, resource losses may not be easily compensated, posing a challenge for pollinator persistence and network stability.

Simulation studies projected a shift in plant species distributions up to several hundred kilometres northward[7,31], with forests likely to shrink in the South and expand to the North. An uphill shift is also predicted in Europe[7,16–18], while about 60% of mountain species face local or global extinction[14]. Many European species will struggle with the changing climate[18], with Southern Europe likely to lose numerous species, while Northern regions may lose boreal but gain Euro-Siberian species[14]. Southern Europe might experience a species migration from Northern Africa or other more arid areas. Although this is likely, we cannot infer such trends and consequences from our data. Nonetheless, such plants need to be considered as invasive alien species, which might negatively increase stress on local plants by competition or could beneficially fill vegetation gaps left by the native species[15,32]. This may coincide with the arrival and establishment of potentially invasive alien bees, such as the recent appearance of *Apis florea* in Malta[33], leading to increased competition for resources. In addition, these can introduce diseases to which native bees have little or no resistance, further threatening their populations[33].

Regarding temporal shifts, advancements in spring phenology have been observed in the Northern Hemisphere of around 2.5 days per decade[30,34]. It is well known that a shift in flowering time is mostly dissociating for plant-insect pollinator interactions[30], and this decoupling will lead to unknown implications for biodiversity and ecosystem function[23,24]. Our empirical results indicate that Southern Europe will experience low resistance and resilience in summer, meaning that a probable flowering shift is less likely, due to fewer plants flowering in autumn. On the other hand, Northern Europe will experience lower resistance and resilience in late spring/early summer (Supplementary Fig. 9).

Our results indicate that throughout the EU, honey bees have few major food resources (Supplementary Fig. 2) yet maintain a diverse range of collected pollen (Fig. 1), likely to ensure a broad nutrient spectrum[19,20] and to dilute toxic plant secondary metabolites[35]. A poor diet will make bees especially vulnerable, as climate change may increase parasite and disease threats[19,36,37]. The decline in floral resources, as projected in our climate scenarios, due to plant losses or shifts in geographic and temporal distributions, will make it increasingly difficult to maintain pollen diversity. This challenge could lead to malnutrition and compromise bee health[19,36]. Besides diversity, lower resource availability might lead to lower colony efficiency, size, and offspring recruitment. Our study showed that most of the major relevant species are not well adapted to high temperature or low precipitation regimes (Supplementary Fig. 10), with a potential strong reduction of their abundance in such scenarios. It was found that a "phenological gap"[38,39] of 15 days of insufficient floral resources severely impacts colony development[38]. Several studies have identified a food deficit for honey bees in June and July[40,41], and our results suggest that rising temperatures across Europe will further exacerbate this phenomenon.

Despite the challenges of future climate change, the honey bee is an efficient, polylectic forager adapted to diverse geographical and climatic conditions in the wild and managed beekeeping environments[21]. While we recognise that floral preferences and foraging strategies differ between pollinator species, most bee species do forage on plants overlapping with such used by honey bees[42]. Therefore, the patterns of floral resource change we identified will be ecologically relevant for broader agroecosystem functioning, even if species-specific responses vary. The trends shown here will probably have an even stronger impact on wild bee and other pollinator species, particularly those exhibiting specialistic foraging patterns (as e.g., mono- or oligolecty, long-tongued bees), restricted geographic distributions, and short foraging windows[19,21]. However, more studies are needed in this regard. Furthermore, the phenological gap will probably impact solitary bees more than the social honey bee since the latter can store pollen and honey to survive periods of dearth[39,43]. Some plants might shift northward or uphill, as previously stated, but pollinators may not follow, as described in certain bumblebee species[44]. Although most studies have concluded that shifts in pollinator phenology have been keeping pace with advances in flowering, there is, however, a high risk of asynchrony range shifts and a consequent decoupling between plants and pollinators with temperature increase[24,30,45].

In summary, our study highlights that climate change will profoundly impact bee foraging resources. Many plants are already at or near critical limits, disrupting their availability and reducing floral resource diversity across Europe in the short term. Southern Europe faces the most severe effects, with diminished plant resources and greater risks to bees, while Northern regions also experience notable impacts. These changes challenge honey bees' ability to maintain a diverse pollen diet, potentially leading to malnutrition and reduced colony efficiency. The broader consequences include potential declines in both plant and bee populations, stressing the urgent need for adaptive and rapid strategies to mitigate such declines, such as the UN Decade of Ecosystem Restoration, which aims to protect and revive worldwide ecosystems by stopping their degradation. Addressing these challenges is crucial for preserving biodiversity and ensuring food security in the face of ongoing climate change.

## Methods
### Pollen sampling
Pollen samples were collected simultaneously every two weeks between May 1st and August 27th 2023, from 310 stationary apiaries distributed across all 27 European Union (EU) countries. These pollen samples were voluntarily collected and provided by participating beekeepers from their own managed honey bee colonies, solely for scientific research and under written consent. Sampling was organised through national coordinators in each EU member state, who served as contact points between the INSIGNIA-EU consortium [https://www.insignia-bee.eu] and participating beekeeper citizen scientists and ensured adherence to standardised sampling procedures. Sampling was performed by activating standard pollen traps for approximately 24 hours on two hives per apiary during routine beekeeping practice. As domesticated and managed *Apis mellifera* colonies and their hive products are not subject to wildlife collection regulations or national access restrictions within the EU, no governmental sampling permits were required. No animals were harmed during pollen collection. All publicly shared data were anonymised in accordance with GDPR and with the agreement of the contributing beekeepers. This included removal of beekeeper identifiers, addresses, and any other personal information, as well as truncation of geographical coordinates in public metadata. Anonymised consent documentation for participating citizen scientist beekeepers is available from the authors upon request.

About 5 g of pollen per apiary (an equal homogeneous amount from each hive) was gathered and stored in capped vials together with 12 g of silica gel, following Quaresma et al.[46]. Of the 2 790 expected pollen samples, only 2 500 were analysed. The shortfall of 290 samples was due either to PCR amplification failure, logistic issues, or the absence of collected pollen.

### DNA metabarcoding
Samples were homogenised in a magnetic stirrer by adding 4 ml of sterile ultrapure water to 2 g of silica-dried pollen. A volume of 200 μl of the homogenate solution (~50 mg of pollen per sample) was placed in 1.5 ml tubes and centrifuged at maximum speed for 3 min. After centrifugation, the supernatant was removed. Before DNA extraction, 300 μl of CTAB buffer was added to the 1.5 mL tubes containing the stored pollen. After thorough mixing, pollen samples were transferred to 2.0 ml screwcap tubes containing a custom-made mixture of zirconia beads of varying sizes (100 μm: 80 mg; 200 μm: 38 mg; 400 μm: 85 mg; 800 μm: 97 mg; 3 mm: 2 beads) to ensure efficient

 

rupture of pollen exines across a wide range of grain sizes. Pollen samples were then ground in a Precellys 24 tissue homogeniser (Bertin Instruments) for three cycles at 6 200 rpm for 5 s. DNA extraction was then carried out using the Maxwell® RSC PureFood GMO and Authentication Kit (Promega) with the Maxwell® automatic extractor (Promega), following the manufacturer's instructions. Positive and negative controls were included in each extraction batch.

DNA metabarcoding was performed using the dual-indexing approach following Sickel et al.[22] for the internal transcribed spacer 2 (ITS2) nuclear DNA fragment. The first PCR was carried out in triplicate in a 10 µl total volume containing 5 µl of Q5 High-Fidelity 2X Master Mix (New England Biolabs), 0.5 µl of primers ITS-S2F[47] (5′-ATGCGATACTTGGTGTGAAT-3′) and ITS-S4R[48] (5′-TCCTCCGCTTATT-GATATGC-3′) at 10 µM, and 1 µl of DNA. The PCR cycling profile consisted of 98 °C for 3 min, 35 cycles of 98 °C for 10 s, 52 °C for 30 s, and 72 °C for 40 s, and a final extension of 72 °C for 2 min.

Amplicons were then subjected to a second PCR to incorporate unique indexes into each triplicate separately. This indexing reaction was performed in a total volume of 14 µl, containing 7 µl of KAPA HiFi HotStart ReadyMixPCR Kit (Kapa Biosystems), 1.4 µl of each oligonucleotide (1 µM), and 2.8 µl of each first-round PCR amplicon, previously diluted 1:1. Thermal cycling conditions were 95 °C for 3 min, followed by 8 cycles of 95 °C for 30 s, 55 °C for 30 s, 72 °C for 30 s, and a final extension of 72 °C for 5 min. Indexed amplicons were normalised using SequalPrep™ Normalisation Kit 96 wells (Invitrogen) to ensure equalised libraries. After normalisation, all the samples from each PCR plate were combined into one pool.

The concentration of each pool was assessed using the Epoch Microplate Spectrophotometer (BioTek Instruments). The pools were then combined equimolarly, purified using 0.75× reversible immobilisation paramagnetic beads (Agencourt AMPure XP), and eluted in 50 µl. Final pools were quality-controlled on a TapeStation 2200 using the HS D5000 kit (Agilent Technologies) and a SYBR green quantitative PCR assay using the KAPA Library Quantification kit (Kapa Biosystems) with the QuantStudio™ 5 Real-Time PCR (ThermoFisher Scientific). The sequencing libraries were diluted to 2 nM, spiked with 10% Illumina-generated PhiX control library, and then sequenced on the Illumina MiSeq using 2 × 250 cycles v2 chemistry according to the manufacturer's instructions.

Pools were de-multiplexed in the BaseSpace Sequence Hub based on unique indexes. Raw sequencing data were deposited publicly at NCBI SRA under BioProject PRJNA1198597. Data were processed according to Leonhardt et al.[49] and the pipeline available at https://github.com/chiras/metabarcoding_pipeline. Raw sequencing reads were merged, quality filtered (length <200 bp and > 500 bp, no ambiguous base pairs), dereplicated, denoised, and chimaeras were removed to generate amplicon sequence variants (ASVs). ASVs were directly classified at the species level using global alignments with VSEARCH v2.15.2[50], a threshold of at least 97% sequence similarity (top-hit selection), as in Sickel et al.[22]. Unclassified reads were subjected to SINTAX[51] hierarchical classification, with a threshold of 90%. The ITS2 reference databases used in both taxonomic classification steps were created by Quaresma et al.[52]. Taxa that could not be confidently distinguished at the species but at genus level, i.e., due to recent evolutionary divergence and frequent hybridisation resulting in low interspecific variability in the ITS2 region (e.g., *Brassica napus*, *B. rapa*, and *B. oleracea*), were treated at the genus level only.

## Climate data

Mean monthly temperature and monthly accumulated precipitation were extracted from the high-resolution (0.5°) gridded dataset CRU TS4.08, provided by the Centre for Climate Research at the University of East Anglia[53]. Climate data was only available monthly, resulting in two sampling points per site sharing the same climate metadata.

## Statistical analysis

Data were analysed using the R[54] v4.4.1 software within the R Studio v2024.04.2 + 764 environment[55] using the R packages phyloseq[56] v1.48.0, tidyr[57] v1.3.1, speedyseq[58] v0.5.3.9021, dplyr[59] v1.1.4, viridis[60] v0.6.5, bipartite[61] v2.20, ggplot2[62] v3.5.1, ggsci[63] v3.2.0, gghighlight[64] v0.4.1, rnaturalearth[65] v1.0.1, sf[66] v1.0.16, geosphere[67] v1.5.18, gdm[68] v1.55.0.9.1, scales[69] v1.3.0, foreach[70] v1.5.2, doParallel[71] v1.0.17, doSNOW[72] v1.0.20, progress[73] v1.2.3, patchwork[74] v1.3.0.9000. The R code is available at https://github.com/chiras/HoneyBee-ResistanceResilience. Processed data and metadata of samples are publicly available at this GitHub repository and as well under https://doi.org/10.5281/zenodo.17578272. Metabarcoding data were transformed to relative read abundances (RRAs). Taxa contributing less than 1% to a given sample were excluded to reduce noise and focus on ecologically meaningful contributors to the pollen diet. Multiple ASVs from the same plant species were aggregated into a single taxon to avoid artificial inflation of diversity metrics.

Pollen diversity: For each sample (site x time point), we estimated Shannon entropy ($H'$) and Effective Species Richness ($e^H$), calculated the crop contribution to diets, and visualised spatial and temporal trends across Europe. This allowed us to examine how floral resource diversity, crop type, and composition vary geographically and seasonally, providing a quantitative baseline (Objective 1) against which projected climate-driven changes can be evaluated. To assess large-scale spatial patterns, we fitted an exploratory linear model relating $H'$ to latitude, including latitude as a quadratic term to capture a non-linear gradient, as hypothesised mid-latitude effect. Model diagnostics showed no evidence of heteroscedasticity (Breusch–Pagan $p = 0.24$) and approximate normality of residuals (Shapiro–Wilk W = 0.98), supported by visual inspection. We also fitted month-stratified models to evaluate seasonal variation in the latitude–diversity relationship. $e^H$ and crop contribution were used to visualise patterns across Europe, latitudes, and months.

A non-metric multidimensional scaling (NMDS) using Bray-Curtis dissimilarities was conducted to visualise compositional variation across sites and time points in an unconstrained ordination space. The ordination was performed with five dimensions ($k = 5$), selected based on minimising stress (final stress = 0.098). To quantify the contribution of environmental variables, we applied Generalised Dissimilarity Modelling (GDM), which models compositional turnover as a function of geographic and climatic gradients. GDM allowed us to estimate the proportion of variation in pollen composition explained by site, temperature, and precipitation. To identify dominant pollen sources in the dataset, we calculated the 25 most abundant plant taxa across all samples by summing RRAs. These were visualised as stacked barplots to highlight the relative contribution of key wild and cultivated taxa. This summary aimed to contextualise floral resource composition across Europe (Objective 1), particularly in relation to the reliance on crop species.

Historical developments of temperature and precipitation at the sites: To contextualise the magnitude of projected climate change scenarios (up to +5 °C and −50 mm), we examined historical changes in temperature and precipitation at each of the 310 sampling sites over the past 50 years. This was done to ensure that our future scenarios are ecologically realistic and grounded in empirical trends (Objective 2). While large-scale climate change is well established, confirming site-specific warming and drying trajectories is essential for interpreting regional foraging risks. To this end, we applied Wilcoxon paired tests comparing monthly temperature and precipitation between 1970–1975 and 2018–2023 for each site and calendar month. These non-parametric tests were chosen to accommodate deviations from normality and to provide robust, site-level statistical confirmation of historical climate change. Summaries by latitude and season are shown in Supplementary Fig. 5.

Climatic response profiling of plant taxa: Our dataset spanned much of Europe but was irregularly sampled and not spatially continuous. We therefore restricted risk assessments to species at sites where they were directly observed, applying an empirical exceedance analysis. Modelling approaches (e.g., SDMs) were not pursued because they typically rely on spatially continuous covariate layers and denser, raster-like coverage to extrapolate across unsampled areas, which are conditions not met here with our point-local data. In preparation for such risk assessment, we estimated climatic response profiles for each plant taxon by pairing the empirical association between pollen abundance and site and date-level temperature and precipitation (Objective 3). RRA of each species was paired with the corresponding climate values at each sampling location and date. These values were then used to construct non-parametric kernel density estimates (KDEs), yielding species-specific distribution-free approximations of their realised climatic niche envelopes. These KDE profiles reflect the abundance-weighted climatic associations of each species, without assuming a parametric distribution or functional form. These profiles were then used in an exceedance analysis to assess risk under future climate scenarios (see below). To ensure robustness, only species with data from >5 unique sites and that occurred in at least 10 different samples were retained. As illustrative examples, temperature and precipitation response curves are shown for the 25 most abundant taxa in Supplementary Fig. 10.

Exceedance-based floral resource risk assessment: For each site and sampling time point, we calculated the percentile-based risk intervals of the site-specific temperature and precipitation within the empirical KDE-based climatic response curve of each plant taxon (Objective 4). This analysis was applied using both current and projected climate conditions. Projections included incremental warming scenarios (0 to +5 °C in 0.5 °C increments) and precipitation reduction scenarios (0 to −50 mm per month in −5 mm steps), for each site and time point. Plants were classified into risk categories based on these percentile thresholds: for temperature, values ≤90th percentile threshold were considered low risk, >90th moderate risk, >95th high risk, and >99th critical risk; for precipitation: values ≥10th percentile threshold were low risk, <10th moderate risk, <5th high risk, and <1st critical risk. We then aggregated the RRAs of plants in each risk category per site and time point. This exceedance-based approach allowed us to estimate the proportion of floral resources that may experience climate conditions outside the observed range of their realised niches at each site and date. This method is transparent, data-driven, and conservation-relevant. Conceptually, it aligns with frameworks such as the IUCN Red List, which applies threshold-based criteria to assess ecological stress and extinction risk. These proportions were summarised by latitude to examine broad geographic trends. Means per latitude were visualised using a composite bar plot, showing the effects of increasing temperatures and reduced precipitation across different latitudes. We were further interested in how resource risks change over the sampling season, so we created graphs showing the proportion of pollen resources classified as at least "moderate risk" using the classification described above along the temperature and precipitation gradient at different latitudes. Graphs are shown in the main text only for a subset of scenarios, while all scenarios are visualised in the Supplementary Information. Finally, we fitted a multi-factorial linear model to test how the proportion of floral resources at risk was affected by climate change scenario, latitude, sampling month, and their interactions. This allowed us to evaluate whether projected risks varied systematically with geography and season.

Resistance and resilience: To better understand the potential for ecological persistence or recovery under climate change, we quantified resistance and resilience of floral resource communities in response to combined temperature and precipitation stress (Objective 5). We conducted a combined exceedance analysis of precipitation and temperature by calculating, for each sample and scenario combination, the proportion of species classified as at risk according to the above definitions due to either precipitation reduction or temperature increase. Resistance was defined as the proportion of non-risk taxa in the community, computed as:

$$Resistance = 1 - \frac{S_{risk}}{S_{no-risk} + S_{risk}} \quad (1)$$

where $S_{risk}$ is the number of taxa in the sample classified as at least "moderate risk" under the given scenario, and $S_{no-risk}$ is the number classified as "low risk".

To assess resilience potential[75,76], we identified replacement taxa that could hypothetically compensate for taxa at risk ($S_{risk}$), assuming range shifts or phenological substitution across space or time. A taxon was considered a suitable replacement for a given at-risk species if it met the following criteria: (1) occurred within a 500 km radius of the focal sample (i.e., geographically proximal); (2) was classified as "low risk" for both temperature and precipitation under the same climate scenario; (3) belonged to the same plant family, serving as a proxy for morphological and nutritional similarity and for maintaining phylogenetic representation; (4) had not already been selected as a replacement in that sample, to avoid redundancy and maintain taxonomic diversity. For each plant taxon in $S_{risk}$, we searched for suitable replacement candidates across the full dataset. Resilience was then defined as the proportion of at-risk taxa for which at least one suitable replacement taxon was identified, calculated as:

$$Resilience = \frac{S_{replacement}}{S_{risk}} \quad (2)$$

where $S_{replacement}$ is the number of at-risk taxa for which replacements were available under the criteria above.

### Reporting summary
Further information on research design is available in the Nature Portfolio Reporting Summary linked to this article.

## Data availability
The raw sequencing data generated in this study have been deposited in the NCBI SRA database under accession code PRJNA1198597. The processed sequencing data (for process see code availability), climate data, and metadata of samples are available at Zenodo 10.5281/zenodo.17578272 [https://doi.org/10.5281/zenodo.17578272][77] and GitHub chiras/HoneyBee-ResistanceResilience [https://github.com/chiras/HoneyBee-ResistanceResilience].

## Code availability
The pipeline for processing raw sequencing data for metabarcoding is publicly available at GitHub chiras/metabarcoding_pipeline [https://github.com/chiras/metabarcoding_pipeline]. Code for all downstream analyses is publicly available at Zenodo 10.5281/zenodo.17578272 [https://doi.org/10.5281/zenodo.17578272][77] and GitHub chiras/HoneyBee-ResistanceResilience [https://github.com/chiras/HoneyBee-ResistanceResilience]. All display items presented in the main manuscript and supplementary information can be reproduced from this public data and code.

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

## Acknowledgements

We thank all Citizen Scientists and National Coordinators from the 27 EU countries for their invaluable contributions to pollen sampling. We also acknowledge Maíra Costa, Maria Celenza, and Maria João Caldeira for their assistance with pollen homogenisation and DNA extractions. The Portuguese Foundation for Science and Technology (FCT) supported A.Q.'s PhD scholarship (2020.05155.BD; DOI: 10.54499/2020.05155.BD). FCT provided financial support through national funds to CIMO and SusTEC via FCT/MCTES (PIDDAC): CIMO under UIDB/00690/2020 (DOI: 10.54499/UIDB/00690/2020) and UIDP/00690/2020 (DOI: 10.54499/UIDP/00690/2020); and SusTEC, under LA/P/0007/2020 (DOI: 10.54499/LA/P/0007/2020). This work was conducted within the framework of INSIGNIA-EU: Preparatory action for monitoring of environmental pollution using honey bees (European Union service contract 09.200200/2021/864096/SER/ ENV.D.2).

## Author contributions

A.Q., A.K., and M.A.P. conceived the ideas and designed the methodology. A.K. and A.Q. conducted the analysis. A.Q., A.K., and M.A.P. drafted the manuscript. Particularly A.Q., M.A.P., A.K., and N.C., but all authors contributed to improving the manuscript. J.B., W.B.B., and I.R. modelled the apiaries' location. R.B., K.G., F.H., O.K., M.A.P., I.R., F.V., N.C. and JvdS designed the pollen sampling methodology, prepared all the materials and manuals for pollen collection by Citizen Scientists, and obtained the INSIGNIA-EU funding. All the authors critically reviewed the manuscript for important intellectual content.

## Funding

## Competing interests

The authors declare no competing interests.
