## [Peer Review file · Nature Communications]

Honey bee food resources under threat from climate change

Corresponding Author: Professor Alexander Keller

Version 0:

Reviewer comments:

Reviewer #1

(Remarks to the Author)

The article studies the effects of climate change on honey bee-plant interactions. This is done using a huge dataset of pollen samples that were collected by honey bees in a host of European countries. The pollen were identified using DNA metabarcoding. The authors applied various data analyses to conclude that climate change will negatively affect foraging resources of honey bees. A R-based pipeline accompanies the article, which I greatly appreciate as it facilitated the review work greatly (despite the somewhat unhandy sequential nature of said pipeline).

In the introduction, the authors state that they selected honey bees as their model organism, which is fine. However, the statement “this makes it an ideal species for assessing the broader impacts of climate change on plant-pollinator interactions” is overly general. I would like to point out (having some, but very limited, experience with community datasets of pollinator-plant interactions) that most bee species exhibit wildly different (foraging) behavior from honey bees. Thus, any general statements as to the broader impacts of the study should be limited to social, domestic, bees. Honey bees are not a good model organism for social wild bees, and not at all for solitary wild bees. That does not take away from the importance of the study, honey bee pollination is critical for society, and understanding how this might change in the future is thus also vital.

I am a statistical ecologist specializing in statistical methods for multivariate analysis, and so I will mostly focus on the data analyses, while (mostly) refraining from the ecological conclusions from the authors. Another reviewer is more suitable to comment on those aspects of the article. The authors should be aware that there is an ongoing shift in statistical community ecology in moving away from “traditional” methods used for data analyses, and more towards model-based approaches. In this debate, I am on the model-based “side”, which leads me to being overly critical on the motivation for and execution of the data analyses in the article.

In the “main text” the authors note that “to assess potential risks to pollination networks, they examined climatic niche distributions of the plant species in the pollen samples, while estimating the likelihood of interaction losses under climate change scenarios”. When diving further into the methods, the authors categorize their analyses in 1) pollen diversity, 2) historical developments of temperature and precipitation, 3) plant temperature and precipitation niches, 4) food resource risk assessment, and 5) resistance and resilience. These analyses by all accounts in line with most of what most of community ecology does. Unfortunately, most of community ecology poorly understands the motivation for applying certain analyses, and from a statistical standpoint, most of the analyses in community ecology have little merit. I continue my review by the same categorization that the authors followed.

I note that most of the analyses are insufficiently motivated with respect to the goal that the authors outline in the main text: what questions, subgoals, or otherwise do these analyses answer? I urge the authors to carefully consider how each analysis or method contributes to the goal of the paper, and elaborating this in the article. Some of the details necessary to understand what is done in the analyses are actually covered in the “results” section, while the “methods” section omits large amounts of detail.

1) Pollen diversity

L491-2: “Effective species richness was estimated plotted on a map”, why? Which goal does this serve or which question does it answer?

L493: “We fitted a quadratic model of effective species richness explained by latitude”, why? What kind of quadratic model? What assumptions were checked to make sure that the results were valid? Does the model predict effective species richness

well or did the authors not consider its predictive accuracy at all? I suspect the latter.

L493-495: "A NMDS was conducted to determine the effect of temperature and precipitation" I fail to see how an NMDS contributes to this effect: NMDS is an unconstrained ordination, and only post-hoc can the axes (which are unitless and rotationally variant) be "tested" for the "effect" of precipitation and temperature. Here, "testing" means to check if the scores of the axes (i.e., in an ordination plot) are structured in some way: increasing in temperature from left to right or decreasing in precipitation from top to bottom, for example, but in no way can the authors from this conclude that pollen diversity is affected by these environmental variables.. The authors fail to clarify the number of axes in the NMDS (usually determined in terms of stress), while the two used axes might very poorly represent the authors' data. Quantifying axis relevance in NMDS is not a trivial endeavour (unlike in other classical ordination methods that come with accompanying eigenvalues), though it is generally done by looking at the "stress" measure that NMDS minimizes. Regardless, a more careful consideration of the number of axes would not have made this a more convincing analysis. After all, NMDS is not a method for estimating effect of environmental variables, this is done using constrained ordination methods (RDA or CCA) or more advanced statistical methods such as vector GLMs, reduced rank vector GLMs (i.e., model-based constrained ordination) or generalized linear latent variable models (model-based ordination and JSDMs generally).

- L495-497: "GDM was used to estimate how much of the pollen composition was explained by site, temperature, and precipitation". Here, the authors are admitting using a tool to get a metric out, which (by all accounts) is considered by statisticians to have little merit for scientific importance. If the authors so wished, the same metric could have been calculated from their post-hoc analysis of NMDS scores, but instead the authors applied another method, which could have been entirely avoided.

- L497-498: Barplots were made for summarizing the most abundant pollen. Why? What goal do these serve?

2) Historical developments

The authors used a Wilcoxon paired test to test if the mean temperature and precipitation of the two time windows has changed. Personally, I think this is a well accepted fact in science, and thus again do not see the added value of this test to the study.

3) Plant temperature and precipitation niches

L508-509: "Temperature data were rounded to 0.5°C, and precipitation data were square-root transformed due to a skewed distribution". Why did the authors round the temperature data, and why is a skewed distribution for the precipitation data a problem? I worry that the authors do not understand the assumptions that are made with their analyses, how to check the consequences of these assumptions, and consequences (or lack thereof) of violating certain assumptions (such as normality).

L509-512: "Temperature and precipitation niche distributions for each plant species were estimated as normal distributions given the mean and standard deviation of temperature and precipitation, respectively, weighted by the abundance data over all samples."

All details with respect to the analyses are omitted: one can only understand what the authors have done by digging into the accompanying github repository, because generally the article omits so much detail that it would be impossible to understand, let alone to reproduce, without the accompanying code.

From the text, "estimate" would indicate to me that the authors fitted a regression of some sort (the reference to transformed skewed data would imply a linear regression), but the code makes clear that the authors do not fit a model at all. What the authors do takes some time to figure out, as the code is not extensively commented throughout. The authors calculated the sample mean and sample variance of the relative abundances of plant species, but how these constitute valid estimates of "distribution" is entirely unclear to me (i.e., a statistical ecologist that frequently works for methods of species distributions). The only product in the article in terms of "distribution" that I can find is Figure 1A, and Extended data Figure 10. Figure 1A plots effective richness on a map, and thus does not constitute a predicted distribution of any kind.

Extended Data Figure 10 is also not a "distribution", but it is a response curve, despite the fact that the authors call it a distribution. In Extended Data Figure 10 the sample mean and sample variance of summed relative abundances are used to plot a Gaussian kernel, without any regard for the accuracy of the response curve (since it is not a distribution, while response curve might be an apt name for what is shown here), and how it is relevant for the original data. The authors cannot make statements about this, as no model has been fitted whatsoever.

This can be seen as an attempt at a linear regression, where temperature and precipitation are separately used as response variables, and "accumulated species" is used as a categorical variable, thus estimating the mean of temperature and precipitation per species, but while allowing for heterogeneity of variances per "accumulated species". Making that connection exposes multiple deficits with the authors' approach: 1) the code could be much more concise, 2) a normality assumption for precipitation is not realistic: precipitation cannot be negative, 3) a skewed error distribution is not a problem, the authors are only interested in prediction, and violation of the normality assumption only affects the statistical uncertainty in a linear regression (and the authors are not quantifying statistical uncertainty at all). In fact, by using the Wilcoxon signed-rank test on temperature and precipitation the authors already account for the non-normality effect on test results, 4) the authors are not analysing their relative abundances: they are analysing temperature and precipitation while stating that they are analysing the species data, i.e., the authors are confusing the relationship of y and x: temperature and precipitation affects the abundance of species, and the distribution of species is predicted using temperature and precipitation. Here, the authors are using species abundances to calculate the (mean) temperature or precipitation of sites where species were present, which effectively ignores nearly all information in the pollen data. Although the x and y relationship can be reversed without consequence when there is only one variable impacting the relative abundance of pollen at sites in a linear regression, this is not true when we assume that there is more than one variable affecting the pollen, as is the case here. Further, there are most likely confounders that affect pollen distribution, which are also not accounted for in this analysis. In

summary, the collinearity of precipitation and temperature is very likely to impact the results, and separate linear regression cannot be fitted to the data while expecting accurate results.

4) Food resource risk assessment

As noted, the calculation of distributions is inadequate, and 4) is based on 3), so that 4) too needs a considerable rework. Further, note that in L517-518 the authors write “the quantile was calculated”, but there is no such thing as “a quantile” as a quantile defined is a range, the word does not take meaning until that range is defined. This is in contrast to percentiles.

In this part of the analysis, the authors fit a linear model and perform model selection using AIC, after which statistical significance is assessed. I have two notes on this: 1) stepwise model selection has considerable issues that the authors should address, details on this can be found widely in the (ecological) literature. The debate is not “hot” anymore, and AIC model selection has largely been put on the back-burner due to its widely recognized flaws, 2) model selection by information criteria cannot be combined with the interpretation of statistical significance. Omitting variables by information criteria naturally leads to better fitting models, and thus artificially increased statistical significance (inflated Type I error). This is commonly referred to as “p-hacking”, and is considered a statistical sin. The two paradigms: information criteria and testing (exploratory and confirmatory), are not unifiable.

5) Resistance and resilience

This calculation depends on the previous analyses with considerable flaws, and thus also needs a rework. At the same time, the equation the authors provide in the text is not referenced, so determined whether or not this is a sensible estimate of resistance (or resilience) is entirely unclear.

Overall, I conclude that the analyses in the article lack both statistical support and ecological motivation. I would have personally reanalysed the data, but due to the sequential nature of the pipeline, and the lack of intermediate products stored, I could not find sufficient time to do the analysis. My recommendation is for the authors to completely redo their analyses, as it can be done considerably more succinctly, and because I expect various (if not all) of the results to change. I recommend that the authors dive into the world of multivariate statistical models, as at least three, if not four, of their analysis can be made redundant by fitting a single vector GLM (as in the `mva` R-package) or single GLLVM (as in the `gllvm` R-package). The compositional nature of the data can be addressed by appropriately selecting the model structure: the total read count should be included as an offset, and the response distribution should be one of Poisson, negative-binomial or zero-inflated versions thereof. I would be happy to provide the authors with more details for a reanalysis if needed.

For now, I have refrained from extensive commenting on the text. That is not by lack of interest – the study is interesting and I would like to see an improved version. However, I expect the present version to still undergo significant changes when the analyses have been improved, so that commenting on (e.g.) the results is not relevant at this stage.

Bert van der Veen, 14-03-2025

(Remarks on code availability)

The code is available, and can be run without (much) further intervention, on which the authors have done a reasonable good job.

The code is not well commented throughout, so that working through the code, and understanding what the authors did, took a considerable amount of time, and was challenging. Ideally, this should be improved. Having said that, it would not have been possible to understand what the authors did in the article, without reviewing the code.

Note that in the first “Umbrella” script the authors incorrectly (for anyone else accessing the code) set a working directory. The sequential nature of the pipeline, without storage of and reference to intermediate results, makes reanalysis extremely difficult.

The pipeline would function better (for reanalysis) if each analysis was stand-alone.

Reviewer #2

(Remarks to the Author)

The manuscript presented to me is very well written. It was a pleasure to read it, and I found it difficult to point out any major shortcomings. The topic is very significant, and the research is original and important. The methods are suitable, although honeybees are not the best representative of bees generally, no other species would allow for such a whole-season analysis and a sufficient number of samples. The Authors are aware of this and discuss their result accordingly. I only missed one piece of information about the apiaries. Did the Authors use only stationary apiaries for their sampling? Moving apiaries into or close proximity to mass flowering and honey-giving crops is a common practice and may affect the results. The description of the results is very detailed, and the Authors did a good job discussing the eventual shortcomings of the climate scenario results. I think it was even a bit too good of a job, and part of these descriptions/interpretations of the received results should be moved to the Discussion, which is short and lacking these interpretations.

I also found a few minor points, which could be improved. In Fig. 1A. the sampling points are somewhat outside the map. I understand that it is because some samples were coming from islands like Lesvos and Cyprus? Can you change the map

underneath to show not just the EU countries but simply all the countries? It would make it easier for the reader to locate the most southeastern sample locations correctly.

Fig2. Can you divide it into two figures? One showing crops, the other wild plants? Although I understand that it shows the total impact, it is difficult to differentiate visually in this form.

(Remarks on code availability)

Yes, I have checked the URL however, I don't feel competent to review it.

Reviewer #3

(Remarks to the Author)

This is a research paper reporting findings from a 4 month study in 2023 across a very impressive number and range of sites in Europe where the authors collected honeybee pollen from apiaries at each of these sites simultaneously every 2 weeks over a 17 week period (presumably). The findings indicate the range of flowering plant species that honeybees visit over May to August at 310 locations, and relate the number to environmental variables. The results provide evidence of the reduced diversity of flowering plants that honeybees are feeding on, and this appears to be related to increased temperature and reduced rainfall. There is no data however on any negative effect on the honeybees, nor any data on forage available at the time of pollen collection. Given the limited range of flowering plants honeybees feed from (seemingly flagged in this paper though framed a little differently), the findings might have major implications for other flowering species and other visiting insects, particularly in light of the fact that we know pollinators are in decline.

I would suggest a re-framing of the use of honeybees as a model, given I believe they do have a preference for a small number of highly abundant flowering plants, as flagged in this study.

There is some lack of clarity and minor suggestions for changes below.

Line 23+ Abstract is missing elements that would be useful, re the dataset, timing, to make it clear that this was one year. This is modelling based on scenarios.

Line 38 Some odd phrasing, e.g. how the word ecology is used in the first sentence, I also question putting value on different ecological interactions.

Line 66 requires a citation. Honeybees are generalists but by focusing on this group you miss the deeper flowers that long tongued groups depend on, and the species that have few inflorescences which I think needs mentioning. Managed honeybees collect pollen which can be very easily located and collected making this a useful group to use to investigate. What's more, your findings further suggest that honeybees feed on a small number of presumably abundant flowering plants. This to me might make the unknown element, the impact of the predicted changes in your analysis on flower visiting insect populations alarming as you mention in the discussion. But flags the importance of monitoring impacts on other species, in addition to an understanding of what is available at these locations at the time.

Line 85 it isn't clear to me whether latitude and temp and/or rain are correlated which is somewhat critical for the interpretation?

Line 74 2500 in total, simultaneously every 2 weeks from 310 apiaries, one apiary per location? May-August in 2023 is that then 8 sampling dates? Which is 2480 samples. This could just be clarified, and could be clearer in the intro and I think should be stated in the abstract at least the timing as this is important.

Line 206 - I'm not certain these results (species richness 5-10) indicates a diverse range of collected pollen, some summary stats would be useful to clarify this and to clarify the proportions of the top flowering plant groups that honeybees are feeding on in the results.

Line 428 why not a specific date?

Line 484 R studio is a platform for analysing data using R software, I think this needs to be referred to appropriately.

Line 482 this indicates pseudo-replication, it's unclear whether this was accounted for in modelling?

(Remarks on code availability)

Reviewer #4

(Remarks to the Author)

The manuscript under review, "Honey bee food resources under threat from climate change," provides valuable insights into the restoration of plant-animal interactions amidst rapid environmental changes and increasing human interference. It effectively conveys the urgency of monitoring and maintaining the balance of species interactions. The manuscript is generally well-written, with a clear and coherent presentation. However, I have a few concerns and suggestions that I

believe could strengthen the study:

Sampling Period: The sampling timeframe appears narrow for drawing conclusions about climate impacts. I recommend extending the study over at least 1–2 additional years to capture interannual variability.

Methodological Comments and Suggestions:

Line 92: It would be beneficial to include a main figure showing plant abundance data stratified by location.

Lines 205–207: Could the authors comment on the occurrence of low-abundance pollen taxa and whether they might result from secondary contamination?

It would be valuable to include a comparison across multiple bee species to assess species-specific preferences in pollen foraging.

Line 438: Please clarify how pollen samples were transferred into 2 ml tubes. Also, include a reference for the zirconia bead size-sorting procedure.

Line 445: Consider placing the Maxwell® 440 RSC PureFood kit information here, and include product details in parentheses.

Line 450: The strategy of maintaining equal DNA concentrations for each sample at the start of PCR is commendable. Did this approach apply to all samples, including those with low concentrations?

Line 454: Please elaborate on how subsamples were selected for indexing. Was pooling involved, and if so, what was the basis? The “1:1 dilution of the first amplicons” requires further clarification.

Pooling Strategy: How were pools equimolarly combined? Please provide methodological detail.

Line 473: The use of 97% sequence similarity—do the authors believe this threshold is sufficient for species-level resolution?

Lines 475–477: The method used to merge undetermined taxa with Brassica is unclear. Please elaborate.

Line 472: Which database(s) were used for taxonomic assignment?

Overall, this manuscript addresses an important and timely topic. With these clarifications and additions, it could make a strong contribution to our understanding of pollinator–plant dynamics under climate stress.

(Remarks on code availability)

Version 1:

Reviewer comments:

Reviewer #1

(Remarks to the Author)

I agree with the authors that various of my concerns have been addressed by careful clarification of the authors' methodologies. The context I attempted to convey in my review, which the authors found misplaced and counter constructive, seems largely be lost on the authors, which is disappointing. I would have liked to see a finer appreciation for the nuances of applied statistical methodologies and statistical best practices reflected in the authors their work, as it would have improved the quality of the manuscript considerably in my perspective. However, I do not think it crucial for final publication in nature communications.

First and foremost, the authors have provided carefully drafted replies to all my comments, which I appreciate. Some of my concerns the authors shared, some of which they did not, which is their prerogative and requires no further addressing. My main concern, that various analyses were not clearly connected to goals or aims of the study, has been carefully addressed, and the manuscript carefully revised, which has greatly improved the readability and transparency of the study.

(Remarks on code availability)

The authors have chosen to keep their pipeline in a sequential fashion, which I believe to be a poor choice. It does not facilitate reproduction or review, albeit the authors believe their study to be in line with open science initiatives. Be that as it may, in line with open science initiative does not correspond to following "best practices", but as the authors have conveyed

unwillingness to change this aspect of their study, there is nothing left to improve in the code.

Reviewer #2

(Remarks to the Author)

The Authors did a good job improving their manuscript. All points raised were corrected or clarified. The manuscript now reads much better and the results are clear.

(Remarks on code availability)

Reviewer #3

(Remarks to the Author)

It looks to me like the authors have done an excellent, thorough job of addressing the comments and amending the manuscript.

(Remarks on code availability)

Reviewer #4

(Remarks to the Author)

Dear authors,

Thank you for the current revision. The manuscript is suitable for publication now. I have no additional comments.

(Remarks on code availability)

Dear authors,

Thank you for the current revision. The manuscript is suitable for publication now. I have no additional comments.

Dear Reviewers,

We sincerely appreciate the valuable and constructive feedback provided on our manuscript. We have carefully addressed all comments, as detailed below, and agree that the revisions have strengthened the manuscript.

Reviewer #1 (Remarks to the Author):

The article studies the effects of climate change on honey bee-plant interactions. This is done using a huge dataset of pollen samples that were collected by honey bees in a host of European countries. The pollen were identified using DNA metabarcoding. The authors applied various data analyses to conclude that climate change will negatively affect foraging resources of honey bees. A R-based pipeline accompanies the article, which I greatly appreciate as it facilitated the review work greatly (despite the somewhat unhandy sequential nature of said pipeline).

>> We thank the reviewer for their thoughtful and constructive comments, which have significantly improved the manuscript. We also appreciate the recognition of the R-based pipeline and acknowledge the limitations of its sequential structure. We have now clarified in the code and repository documentation that key intermediate results are saved during runtime. However, certain steps, such as the generation of plant-specific density functions, require sequential execution and recall in subsequent parts, due to their dependency on in-memory structures, and cannot be modularised.

In the introduction, the authors state that they selected honey bees as their model organism, which is fine. However, the statement “this makes it an ideal species for assessing the broader impacts of climate change on plant-pollinator interactions” is overly general. I would like to point out (having some, but very limited, experience with community datasets of pollinator-plant interactions) that most bee species exhibit wildly different (foraging) behavior from honey bees. Thus, any general statements as to the broader impacts of the study should be limited to social, domestic, bees. Honey bees are not a good model organism for social wild bees, and not at all for solitary wild bees. That does not take away from the importance of the study, honey bee pollination is critical for society, and understanding how this might change in the future is thus also vital.

>> We thank the reviewer for this insightful comment. We agree that honey bees differ substantially in foraging behaviour from wild and solitary bees due to traits such as colony-level recruitment and generalist foraging. Our study does not aim to generalise pollinator behaviour but to assess how climate change may affect floral resources available to *Apis mellifera*, and the resulting nutritional implications. While species-specific responses will differ, honey bees forage on a broad range of plants, many of which are shared with other pollinators. Thus, the patterns of floral resource shifts we report remain relevant for agroecosystem dynamics and may have indirect implications for other bee taxa. We have clarified this scope in the Introduction and Discussion and removed the "model organism".

I am a statistical ecologists specializing in statistical methods for multivariate analysis, and so I will mostly focus on the data analyses, while (mostly) refraining from the ecological conclusions from the authors. Another reviewer is more suitable to comment on those aspects of the article. The authors should be aware that there is an ongoing shift in statistical community ecology in moving away from “traditional” methods used for data analyses, and more towards model-based approaches. In this debate, I am on the model-based “side”, which leads me to being overly critical on the motivation for and execution of the data analyses in the article.

>> We thank the reviewer for their thoughtful remarks and for outlining their perspective on statistical methodology in community ecology. We acknowledge the increasing role of model-based approaches and appreciate their utility. Our study does in fact employ several model-based methods, including Generalised Dissimilarity Modelling (GDM) and multifactorial linear models. That said, we agree that the rationale for specific analyses was not always fully articulated, partly due to prior formatting constraints imposed by a different journal (see reply to reviewer 2), where methods were detached and significantly shortened. We have now revised the manuscript to clarify the motivation for each analytical step.

While we recognize the value of methodological reflection, we respectfully note that general commentary on broad statistical trends, particularly when not accompanied by specific model suggestions or references (e.g., statistical vs. mechanistic frameworks, JSDMs, or vector GLMs), may be better suited to an own perspective piece rather than a manuscript review. We remain confident that our analytical framework is appropriate to the questions posed and the data structure at hand, and we are grateful to the reviewer for helping us strengthen the presentation of our approach.

In the “main text” the authors note that “to assess potential risks to pollination networks, they examined climatic niche distributions of the plant species in the pollen samples, while estimating the likelihood of interaction losses under climate change scenarios”. When diving further into the methods, the authors categorize their analyses in 1) pollen diversity, 2) historical developments of temperature and precipitation, 3) plant temperature and precipitation niches, 4) food resource risk assessment, and 5) resistance and resilience. These analyses by all accounts in line with most what most of community ecology does. Unfortunately, most of community ecology poorly understands the motivation for applying certain analyses, and from a statistical standpoint, most of the analyses in community ecology have little merit. I continue my review by the same categorization that the authors followed.

>> While we acknowledge the comment regarding broader trends in community ecology, we emphasize that our analyses were selected based on their relevance to our research questions and are supported by established methods. We have now clarified our motivations by clearer objectives and more explicitly in the revised manuscript, especially at the end of the introduction.

>> With respect to the reviewer's general statement that "most of community ecology poorly understands the motivation for applying certain analyses," we respectfully note that such broad characterisations fall outside the scope of this manuscript and do not contribute constructively to the review of our work

We also note that none of the other reviewers raised concerns about the suitability or justification of the analyses. Nevertheless, we are grateful for the opportunity to improve the choice, clarity and framing of our methods and hope that the revisions address the reviewer's concerns.

I note that most of the analyses are insufficiently motivated with respect to the goal that the authors outline in the main text: what questions, subgoals, or otherwise do these analyses answer? I urge the authors to carefully consider how each analysis or method contributes to the goal of the paper, and elaborating this in the article. Some of the details necessary to understand what is done in the analyses are actually covered in the "results" section, while the "methods" section omits large amounts of detail.

>> We agree that the connection between our overarching aim and the individual analyses could be more clearly articulated. In response, we have revised the Introduction to explicitly state the five analytical objectives and added brief linking statements at the start of each relevant subsection to clarify how each analysis contributes to addressing the overall research question. We agree, these revisions improve the logical flow and clarify the purpose of each analytical step. Thank you for pointing this out!

1) Pollen diversity

L491-2: "Effective species richness was estimated and plotted on a map", why? Which goal does this serve or which question does it answer?

>> We clarified that mapping effective species richness serves to address Objective 1 by providing a baseline of floral resource diversity across Europe, against which climate-driven changes can be evaluated.

L493: "We fitted a quadratic model of effective species richness explained by latitude", why? What kind of quadratic model? What assumptions were checked to make sure that the results were valid? Does the model predict effective species richness well or did the authors not consider its predictive accuracy at all? I suspect the latter.

>> We thank the reviewer for this important point. The quadratic model was used for exploratory purposes, specifically to examine broad spatial patterns in floral resource diversity (Objective 1), including the hypothesised mid-latitude peak. We have now made this motivation more explicit in the text. As the reviewer correctly notes, this analysis was not intended for prediction, but to describe the geographic structure of diversity across sites and seasons. While we understand that the reviewer may prioritise predictive approaches, we consider this valuable, particularly given that bee foraging diversity has not previously been assessed at this scale. The final model showed no evidence of heteroscedasticity (Breusch–Pagan $p = 0.24$) and approximate normality (Shapiro–Wilk

W = 0.98), supported by residual diagnostics. As noted in the Results, the model's explanatory power was low (adjusted $R^2 = 0.10$), which is consistent with its exploratory purpose.

L493-495: “A NMDS was conducted to determine the effect of temperature and precipitation” I fail to see how an NMDS contributes to this effect: NMDS is an unconstrained ordination, and only post-hoc can the axes (which are unitless and rotationally variant) be “tested” for the “effect” of precipitation and temperature. Here, “testing” means to check if the scores of the axes (i.e., in an ordination plot) are structured in some way: increasing in temperature from left to right or decreasing in precipitation from top to bottom, for example, but in no way can the authors from this conclude that pollen diversity is affected by these environmental variables.. The authors fail to clarify the number of axes in the NMDS (usually determined in terms of stress), while the two used axes might very poorly represent the authors' data. Quantifying axis relevance in NMDS is not a trivial endeavour (unlike in other classical ordination methods that come with accompanying eigenvalues), though it is generally done by looking at the “stress” measure that NMDS minimizes. Regardless, a more careful consideration of the number of axes would not have made this a more convincing analysis. Afterall, NMDS is not a method for estimating effect of environmental variables, this is done using constrained ordination methods (RDA or CCA) or more advanced statistical methods such as vector GLMs, reduced rank vector GLMs (i.e., model-based constrained ordination) or generalized linear latent variable models (model-based ordination and JSDMs generally).

>> We thank the reviewer for this detailed and helpful critique. We fully agree that NMDS is an unconstrained ordination method and does not allow for direct inference about environmental drivers. We acknowledge that our original phrasing was imprecise, and have revised the manuscript to clarify that NMDS was used solely to visualise compositional dissimilarity among samples, not to assess environmental effects. The stress value (0.098) is now included in the Supplementary Figure legend, and the number of NMDS axes ($k = 5$) is clarified in the Methods.

L495-497: “GDM was used to estimate how much of the pollen composition was explained by site, temperature, and precipitation”. Here, the authors are admitting using a tool to get a metric out, which (by all accounts) is considered by statisticians to have little merit for scientific importance. If the authors so wished, the same metric could have been calculated from their post-hoc analysis of NMDS scores, but instead the authors applied another method, which could have been entirely avoided.

>> Regarding the use of Generalized Dissimilarity Modelling (GDM), we respectfully disagree with the characterisation that it lacks scientific merit. GDM is a well-established model-based approach designed to quantify spatial and environmental turnover in species composition. Unlike NMDS, it allows explicit modelling of dissimilarity as a nonlinear function of geographic and environmental gradients, and is widely used in biodiversity, macroecology, and biogeography. We selected GDM precisely because it accommodates

complex species–environment relationships without relying on assumptions of linearity or normality, and because it is well suited to large-scale, high-dimensional community datasets such as ours. We believe this makes it more appropriate than either constrained ordination or regression on NMDS scores, given the structure and scope of our data. We have revised the manuscript to make the distinction between NMDS and GDM clearer.

L497-498: Barplots were made for summarizing the most abundant pollen. Why? What goal do these serve?

>> We thank the reviewer for raising this point. The barplots and summaries of the most abundant pollen taxa support Objective 1, which is to characterise the composition and diversity of floral resources collected by honey bees across Europe. Identifying dominant taxa (both wild and cultivated) provides essential ecological context for interpreting patterns of diversity, potential vulnerability, and seasonal dynamics. This compositional overview also informs Objectives 4 and 5, which assess climate-driven risks and the potential for resilience. We have now clarified this rationale in the Methods and Results. We also wish to emphasise the value of such large-scale descriptive data, which remain rare despite their importance for understanding foraging ecology at continental scale.

2) Historical developments

The authors used a Wilcoxon paired test to test in the mean temperature and precipitation of the two time windows has changed. Personally, I think this is a well accepted fact in science, and thus again do not see the added value of this test to the study.

>> We thank the reviewer for this comment. While overall climate warming is well established, regional and even more local patterns can vary significantly. The included the Wilcoxon paired tests were to confirm that such changes are detectable at the scale of our actual sampling locations. This ensures that the historical context of our study is grounded in site-specific climate trajectories, which is important for interpreting scenario-based projections. In this context, we now adapted the range of simulations for precipitation to reflect a better range of historical developments on local scale. We have clarified this rationale in the revised Methods. We also refer the reviewer to the accompanying figure and caption for further context.

3) Plant temperature and precipitation niches

L508-509: “Temperature data were rounded to 0.5°C, and precipitation data were square-root transformed due to a skewed distribution”. Why did the authors round the temperature data, and why is a skewed distribution for the precipitation data a problem? I worry that the authors do not understand the assumptions that are made with their analyses, how to check the consequences of these assumptions, and consequences (or lack thereof) of violating certain assumptions (such as normality).

>> The reviewer is correct that skewness is not inherently problematic unless parametric assumptions are made. Our initial approach relied on estimating species-specific climatic

means and standard deviations, which assumes approximately normal distributions. This approach followed established works, as e.g.:

*Kühnel, S., Blüthgen, N. High diversity stabilizes the thermal resilience of pollinator communities in intensively managed grasslands. **Nature Communications** 6, 7989 (2015). <https://doi.org/10.1038/ncomms8989>, especially the framework illustrated in Figure 1.*

Rounding temperature values to 0.5 °C and applying a square-root transformation to precipitation were intended to reduce noise and skewness in support of this parametric framework.

However, we recognise that normality may not hold across all species, and our initial, manual validation focused only on the most abundant taxa. In response, we have now replaced the parametric approach with a non-parametric kernel density estimation (KDE) method. This allows us now to estimate plant–climate associations without assuming normality and increases the robustness of our climate risk estimates. In fact, the results and interpretation did not change. New changes are reflected in the revised Methods section and implemented in the updated analysis pipeline.

L509-512: “Temperature and precipitation niche distributions for each plant species were estimated as normal distributions given the mean and standard deviation of temperature and precipitation, respectively, weighted by the abundance data over all samples.”

All details with respect to the analyses are omitted: one can only understand what the authors have done by digging into the accompanying github repository, because generally the article omits so much detail that it would be impossible to understand, let alone to reproduce, without the accompanying code.

From the text, “estimate” would indicate to me that the authors fitted a regression of some sort (the reference to transformed skewed data would imply a linear regression), but the code makes clear that the authors do not fit a model at all. What the authors do takes some time to figure out, as the code is not extensively commented throughout. The authors calculated the sample mean and sample variance of the relative abundances of plant species, but how these constitute valid estimates of “distribution” is entirely unclear to me (i.e., a statistical ecologists that frequently works for methods of species distributions).

>> We thank the reviewer for this close and thoughtful reading. We acknowledge that the original text may have been unclear and thereby led to misunderstandings about both our goals and the methods used. In particular, our use of the term “distribution” was intended in a statistical sense (i.e., a response profile over climate space), but we recognise that it may be interpreted as referring to spatial distributions or even species distribution models (SDMs), which we do not apply. We have now revised the terminology throughout to refer instead to “climatic response profiles” or “abundance-weighted climatic associations.”

>> As the reviewer correctly observes, our analysis does not involve modelling species abundance or presence as a function of climate via regression. In our original approach, we summarised the climatic conditions associated with each plant taxon by calculating abundance-weighted means and standard deviations of temperature and precipitation. However, we agree that this parametric framework assumes normality and may not adequately capture all plants' climatic associations, therefore we switched to the KDE approach as detailed in the previous comment response.

>> These KDE-derived response profiles are then used to assess species-specific climate sensitivity by evaluating whether projected site-level temperature and precipitation values fall outside the observed climatic envelope of each species. This is not a predictive SDM, but a data-driven exceedance analysis designed to highlight the potential for mismatch between future local climates and observed foraging niches. We have substantially revised the Methods section to clarify each step of this procedure and ensure that it is understandable and reproducible without referring to the source code.

The only product in the article in terms of “distribution” that I can find is Figure 1A, and Extended data Figure 10. Figure 1A plots effective richness on a map, and thus does not constitute a predicted distribution of any kind.

>> We have clarified that Figure 1A shows observed, not predicted, effective species richness across sites.

Extended Data Figure 10 is also not a “distribution”, but it is a response curve, despite the fact that the authors call it a distribution.

>> We agree that the term “distribution” may lead to confusion with species distribution models (SDMs) rather than statistical representations. We now refer to the curves shown in Supplementary Figure 10 as empirical response curves not to be confused with fitted or theoretical distributions. Terminology has been updated throughout the manuscript to improve clarity.

In Extended Data Figure 10 the sample mean and sample variance of summed relative abundances are used to plot a Gaussian kernel, without any regard for the accuracy of the response curve (since it is not a distribution, while response curve might be an apt name for what is shown here), and how it is relevant for the original data. The authors cannot make statements about this, as no model has been fitted whatsoever.

>> We thank the reviewer for this comment and appreciate the opportunity to clarify. In the original version, Supplementary Figure 10 displayed Gaussian curves based on abundance-weighted means and standard deviations, following a parametric framework used in previous works (e.g. Kühnel & Blüthgen 2015, *Nat. Commun.*). However, we agree that this approach imposes assumptions that may not hold valid across all taxa, was only verified manually for a subset, and does not constitute a model fit. In response to the reviewer's concern, and in line with above mentioned changes, we have replaced the Gaussian curves following with non-parametric kernel density estimates (KDEs), which more faithfully represent the empirical climate–abundance associations of each species

without assuming normality. These KDEs directly reflect the response curves used in the risk analysis, and the figure is intended solely for descriptive illustration of the 25 most abundant taxa. We have updated the figure, caption, and main text.

This can be seen as an attempt at a linear regression, where temperature and precipitation are separately used as response variables, and “accumulated species” is used as a categorical variable, thus estimating the mean of temperature and precipitation per species, but while allowing for heterogeneity of variances per “accumulated species”.

Making that connection exposes multiple deficits with the authors’ approach: 1) the code could be much more concise,

>> We have revised the relevant sections in the code to improve clarity and conciseness. We apologize that there were still remnants and variables of older workflows not used anymore and might have caused confusions later. Such lines were removed.

2) a normality assumption for precipitation is not realistic: precipitation cannot be negative,

>> We agree and have replaced this method with a non-parametric, kernel density-based approach that does not assume any specific distributional form. This change improves robustness, especially for skewed or bounded variables like precipitation.

3) a skewed error distribution is not a problem, the authors are only interested in prediction, and violation of the normality assumption only affects the statistical uncertainty in a linear regression (and the authors are not quantifying statistical uncertainty at all). In fact, by using the Wilcoxon signed-rank test on temperature and precipitation the authors already account for the non-normality effect on test results,

>> We agree and note that our revised kernel density-based approach no longer relies on distributional assumptions. To clarify, the Wilcoxon test was used only for historical climate trends at sampling locations, not for plant response data.

4) the authors are not analysing their relative abundances: they are analysing temperature and precipitation while stating that they are analysing the species data, i.e., the authors are confusing the relationship of y and x: temperature and precipitation affects the abundance of species, and the distribution of species is predicted using temperature and precipitation. Here, the authors are using species abundances to calculate the (mean) temperature or precipitation of sites where species were present, which effectively ignores nearly all information in the pollen data. Although the x and y relationship can be reversed without consequence when there is only one variable impacting the relative abundance of pollen at sites in a linear regression, this is not true when we assume that there is more than one variable affecting the pollen, as is the case here.

>> We thank the reviewer for this detailed comment and appreciate the opportunity to clarify. We believe this reflects a misinterpretation of our analytical framework. We are

not treating temperature or precipitation as response variables, nor do we model species abundance or distribution using regression techniques. Our approach is as follows:

- For each plant species, we construct an empirical climatic response curve (no longer referred to as a statistical “distribution”), now based on kernel density estimation. These curves represent the species’ realised climatic niche, summarising the temperature and precipitation conditions under which the species was observed, weighted by its relative abundance.
- For each site and climate scenario, we compare projected temperature and precipitation to these species-specific curves to assess whether future conditions fall outside the observed climate envelope. Species are then assigned to risk categories based on exceedance thresholds (e.g., >90th, >95th, >99th percentile).
- We then aggregate the relative abundances of species falling into each risk category at a site to estimate the proportion of floral resources potentially vulnerable to climate change.

This is a threshold-based exceedance analysis, not a regression model. Our focus is on the species historically foraged at each site and whether projected climatic conditions are likely to exceed their empirically observed environmental ranges. This logic aligns with widely used conservation risk assessment frameworks. For example, the IUCN Red List classification system relies on threshold-based criteria to assess extinction risk. These thresholds are not derived from complex models but are applied transparently and directly to empirical data to inform conservation decisions. Our method follows a similar rationale: it uses observed climatic bounds for each species and quantifies site-level risk by assessing the proportion of the community likely to fall outside these bounds under future scenarios. This ensures ecological interpretability and relevance, particularly in policy and conservation contexts.

We reworked the text to make this clearer.

Further, there are most likely confounders that affect pollen distribution, which are also not accounted for in this analysis. In summary, the collinearity of precipitation and temperature is very likely to impact the results, and separate linear regression cannot be fitted to the data while expecting accurate results.

>> We agree that temperature and precipitation may covary, and that additional environmental or ecological factors likely influence pollen composition and plant occurrence. We are aware that ecological responses are complex and multifactorial. Therefore, the following resistance-resilience analysis already jointly considers temperature and precipitation scenarios to evaluate potential buffering through temporal or spatial resource shifts. Such covariation was already noted in the manuscript, especially in the resilience section, and we now further emphasise that our species–climate associations do not capture all potentially relevant drivers. However, our approach avoids the pitfalls of multicollinearity in regression-based analyses by not relying on regression

at all. Instead, we apply a (now non-parametric) abundance-weighted climate profiling method and a threshold-based exceedance analysis.

We also believe that the structure of our dataset supports this approach: our pollen samples are geographically extensive but spatially sparse and not rasterized. Our analysis focuses only on whether a species is likely to experience climatic conditions beyond its empirically observed envelope at the sites where it is currently found, where we can assume constancy in other covariates that would need to be incorporated as variables into predictive modelling beyond the local sites. Attempting to model spatial presence or abundance, such as through SDMs, would be unwarranted given the absence of critical covariates (e.g., land use, soils, agricultural practices) that would substantially affect extrapolation beyond our observed sites. In this context, a simple, abundance-weighted climate profiling approach combined with a threshold-based exceedance analysis, both grounded in empirical observations at local sites, is more defensible than attempting to predict beyond the data.

We hope these rationales and clarifications are now clearly reflected in the revised text.

4) Food resource risk assessment

As noted, the calculation of distributions is inadequate, and 4) is based on 3), so that 4) too needs a considerable rework.

>> As mentioned before, we have revised our method for estimating climatic niches of species to a kernel density estimation (KDE) to empirically model each species' realized niche along temperature and precipitation gradients. KDE avoids assumptions of unimodality and symmetry, and allows us to better reflect skewed or multimodal distributions that are common in ecological data. This is now implemented also in this section.

Further, note that in L517-518 the authors write “the quantile was calculated”, but there is no such thing as “a quantile” as a quantile defined is a range, the word does not take meaning until that range is defined. This is in contrast to percentiles.

>> We appreciate the precision in the reviewer's comment and agree that our original phrasing led to confusion. We have revised occurrences of this issue in the manuscript to “percentile thresholds” or “percentile-based risk intervals”.

In this part of the analysis, the authors fit a linear model and perform model selection using AIC, after which statistical significance is assessed. I have two notes on this: 1) stepwise model selection has considerable issues that the authors should address, details on this can be found widely in the (ecological) literature. The debate is not “hot” anymore, and AIC model selection has largely been put on the back-burner due to its widely recognized flaws, 2) model selection by information criteria cannot be combined with the interpretation of statistical significance. Omitting variables by information criteria naturally leads to better fitting models, and thus artificially increased statistical significance (inflated Type I error). This is commonly referred to as “p-hacking”, and is

considered a statistical sin. The two paradigms: information criteria and testing (exploratory and confirmatory), are not unifiable.

>> We agree with the concerns and have entirely removed the stepwise model selection. In most cases, this did not change final models.

5) Resistance and resilience

This calculation depends on the previous analyses with considerable flaws, and thus also needs a rework. At the same time, the equation the authors provide in the text is not referenced, so determined whether or not this is a sensible estimate of resistance (or resilience) is entirely unclear.

>> The rework is done given the Kernel based approach now. The resistance and resilience calculations are based on transparent and ecologically grounded definitions that quantify the proportion of currently observed taxa likely to persist or be replaced under changing climatic conditions. While simple in formulation, these metrics reflect common conservation-oriented concepts. As the equations are straightforward proportions derived from clearly stated assumptions, we believe external referencing is unnecessary, but we now clarify that these are custom definitions tailored to our risk assessment framework. We however included references that underline the general concepts of resistance and resilience.

Overall, I conclude that the analyses in the article lack both statistical support and ecological motivation. I would have personally reanalysed the data, but due to the sequential nature of the pipeline, and the lack of intermediate products stored, I could not find sufficient time to do the analysis.

>> We respectfully disagree with the reviewer's assessment. Many of the concerns raised appear to stem from misunderstanding our framework (exceedance analysis vs. regression) and earlier ambiguities in our presentation, which we have now addressed through clearer narrative structure, improved terminology, and more transparent documentation of the analytical steps. In the revised manuscript, we have clarified the ecological motivation and statistical rationale for each part of the workflow. Several of suggestions of Reviewer #1 have now been implemented. None of these steps were questioned by the other reviewers.

The full analysis pipeline is reproducible and follows open science best practices. All scripts and data are openly available. Intermediate outputs were in fact stored at most stages of the analysis, and while these files were not reloaded in subsequent scripts, we have now ensured they are consistently named and clearly organised in a dedicated *intermediate.data* folder for transparency and ease of reuse.

My recommendation is for the authors to completely redo their analyses, as it can be done considerably more succinctly, and because I expect various (if not all) of the results to change.

>> We have restructured the entire analytical workflow, now using a kernel density estimation framework to derive species-specific climatic envelopes. This revision affects all downstream analyses. Importantly, despite this substantial methodological change, the main results and interpretations remained consistent. If anything, the updated results indicate more pronounced risk levels than previously estimated, suggesting that our original analysis was conservative. This consistency reinforces the robustness of our findings and the validity of our general approach.

I recommend that the authors dive into the world of multivariate statistical models, as at least three, if not four, of their analysis can be made redundant by fitting a single vector GLM (as in the mvabund R-package) or single GLLVM (as in the gllvm R-package). The compositional nature of the data can be addressed by appropriately selecting the model structure: the total read count should be included as an offset, and the response distribution should be one of Poisson, negative-binomial or zero-inflated versions thereof. I would be happy to provide the authors with more details for a reanalysis if needed.

>> We appreciate the reviewer's suggestion to explore multivariate frameworks such as vector GLMs (e.g., via the mvabund package) and generalized linear latent variable models (GLLVMs). These tools are indeed powerful for specific types of inference, particularly when the goal is to model multivariate responses.

However, our study is conceptually distinct. We do not aim to fit multivariate species-environment regression models or predict abundance patterns. Instead, we aim to assess species-level climatic risk by estimating the likelihood of future environmental conditions exceeding the empirically observed climatic envelopes of currently foraged plant taxa. This approach prioritizes ecological interpretability and transparency over complexity.

From a statistical perspective, methods like GLLVMs require estimating large numbers of parameters across all species simultaneously. Given our dataset (>1,100 taxa across >2,000 samples), these models would face significant challenges related to computational power, overfitting, and identifiability, particularly for rare taxa. Moreover, they introduce assumptions about distributions, link functions, and latent structure that are neither testable in our context nor aligned with our aims.

By contrast, our revised framework uses empirical, abundance-weighted kernel density estimates to characterise species' realised climatic niches, followed by a percentile-based exceedance analysis under climate scenarios. This approach is non-parametric, transparent, and grounded in observed data. It also aligns well with conservation-focused risk assessment frameworks, such as the IUCN Red List, which similarly applies defined threshold criteria (e.g., $\geq 75\%$ loss of a species' population or range experiencing a specified level of decline) to assess levels of threat. Like our method, IUCN assessments prioritise interpretability, data sufficiency, and operational simplicity in support of real-world decision-making. For these reasons, both conceptual and statistical, we chose to retain our current approach, which is well aligned with our study aims and the structure of the data.

For now, I have refrained from extensive commenting on the text. That is not by lack of interest – the study is interesting and I would like to see an improved version. However, I expect the present version to still undergo significant changes when the analyses have been improved, so that commenting on (e.g.,) the results is not relevant at this stage.

>> We understand the reviewer's decision, though we note that the updated analyses, now based on a non-parametric framework, do not violate any potential distributional assumptions questioned. The results remain stable and support the same ecological conclusions, now with increased statistical rigour and greater clarity in exposition.

We believe that part of the reviewer's critique reflects a misalignment of methodological perspective. Our goal is not to model compositional variation via multivariate regression, but to identify ecological risk through exceedance of empirically observed species-specific climatic distributions. This is a different question, requiring a different statistical lens. We hope the revised manuscript now makes this distinction clear and demonstrates that our methods are well suited to our data structure and ecological objectives.

Bert van der Veen, 14-03-2025

>> Thank you for your thorough review and valuable suggestions.

Reviewer #1 (Remarks on code availability):

The code is available, and can be run without (much) further intervention, on which the authors have done a reasonable good job.

>> Thank you

The code is not well commented throughout, so that working through the code, and understanding what the authors did, took a considerable amount of time, and was challenging. Ideally, this should be improved. Having said that, it would not have been possible to understand what the authors did in the article, without reviewing the code.

>> Thank you. We have reworked the entire code in reply to this comment. We included more comments now, also removed parts of the code that were not used anymore and may have been misleading and raised concerns.

Note that in the first "Umbrella" script the authors incorrectly (for anyone else accessing the code) set a working directory. The sequential nature of the pipeline, without storage of and reference to intermediate results, makes reanalysis extremely difficult.

>> We thank the reviewer for pointing this out. In the original code, intermediate results were in fact stored, but all were placed in the base directory. We have now reorganized the pipeline to include a dedicated folder (*intermediate.data*) for storing these files, and we have improved the naming conventions for greater clarity and reproducibility. One major computational step is now omitted if intermediate results are present, and data loaded from file, not recalculated (was previously still executed, although intermediate files were present).

We also note that some steps in the pipeline, such as kernel density estimation, produce R objects (e.g., functions or distributions) that cannot be meaningfully serialized or reloaded across scripts without custom handling. For these components, the current design, where objects are passed between sequential scripts, is necessary for reproducibility and model integrity. We have added explanatory comments in the code to make this logic clearer for external users, and declare which explicit variables are needed downstream. We further specified which dependencies need to be run for individual parts, since in fact not all rely on all others.

The pipeline would function better (for reanalysis) if each analysis was stand-alone

>> As noted above, making each script fully stand-alone is unfortunately not feasible due to the sequential nature of the workflow and dependencies between steps, particularly for complex objects such as kernel density functions and preprocessed matrices. However, as with any R-based workflow, users may store and reload intermediate objects (e.g., via .RData files) at any point. This provides flexibility for reanalysis or adaptation without restructuring the analytical logic.

Reviewer #2 (Remarks to the Author):

The manuscript presented to me is very well written. It was a pleasure to read it, and I found it difficult to point out any major shortcomings. The topic is very significant, and the research is original and important. The methods are suitable, although honeybees are not the best representative of bees generally, no other species would allow for such a whole-season analysis and a sufficient number of samples. The Authors are aware of this and discuss their result accordingly.

>> Thank you for your positive appreciation and valuable comments, which helped us improve the manuscript and enhance its clarity. We also discuss differences between honey bees and other bees now in more detail, also due to comments of other Reviewers.

I only missed one piece of information about the apiaries. Did the Authors use only stationary apiaries for their sampling? Moving apiaries into or close proximity to mass flowering and honey-giving crops is a common practice and may affect the results.

>> Yes, all apiaries were stationary, and we have now included this information in the Methods section. Please see lines 283-285.

The description of the results is very detailed, and the Authors did a good job discussing the eventual shortcomings of the climate scenario results. I think it was even a bit too good of a job, and part of these descriptions/interpretations of the received results should be moved to the Discussion, which is short and lacking these interpretations.

>> Thank you for your comment. You are absolutely right, the two sections are unbalanced, due to much of the discussion being embedded within the Results. Before submitting to *Nature Communications*, we initially submitted to *Nature* which required a different manuscript format. During the reformatting process for *Nature Communications*, this imbalance was introduced, and we apologize for that. Since *Nature*

Communications allows for a combined Results and Discussion section, we opted to merge the two sections in order to preserve as much as possible the structure and content of the version already evaluated by the reviewers.

I also found a few minor points, which could be improved. In Fig. 1A. the sampling points are somewhat outside the map. I understand that it is because some samples were coming from islands like Lesvos and Cyprus? Can you change the map underneath to show not just the EU countries but simply all the countries? It would make it easier for the reader to locate the most southeastern sample locations correctly.

>> We appreciate the reviewer's observation. Rather than expanding the map to include all countries, which would not resolve the issue due to projection constraints, we clarified the sampling locations by explicitly naming the relevant islands in the figure caption. The revised caption now reads: "The map displays the spatial distribution of pollen samples collected across the 27 EU countries, including Cyprus and Malta, as well as the French island of Corsica and the Greek islands of Chios, Crete, Lesvos, and Samos, with colour representing Effective Species Richness (eH)..."

Fig2. Can you divide it into two figures? One showing crops, the other wild plants? Although I understand that it shows the total impact, it is difficult to differentiate visually in this form.

>> Thank you for this valuable suggestion. We have updated Figure 2 and legend accordingly. The revised figure retains a unified layout but now mirrors crop and wild taxa around zero: crop-related impacts are shown on the left (negative axis), and wild plant impacts on the right (positive axis). This preserves the total view while making the distinction between crop and wild components clearer and more intuitive. Also the Figure was updated using color-blind friendly colors according to journal policy.

Reviewer #2 (Remarks on code availability):

Yes, I have checked the URL however, I don't feel competent to review it.

Reviewer #3 (Remarks to the Author):

This is a research paper reporting findings from a 4 month study in 2023 across a very impressive number and range of sites in Europe where the authors collected honeybee pollen from aviaries at each of these sites simultaneously every 2 weeks over a 17 week period (presumably). The findings indicate the range of flowering plant species that honeybees visit over May to August at 310 locations, and relate the number to environmental variables. The results provide evidence of the reduced diversity of flowering plants that honeybees are feeding on, and this appears to be related to increased temperature and reduced rainfall. There is no data however on any negative effect on the honeybees, nor any data on forage available at the time of pollen collection. Given the limited range of flowering plants honeybees feed from (seemingly flagged in this paper though framed a little differently), the findings might have major implications for other

flowering species and other visiting insects, particularly in light of the fact that we know pollinators are in decline.

>> Thank you for your positive appreciation and valuable comments, which helped us improve the manuscript and enhance its clarity.

I would suggest a re-framing of the use of honeybees as a model, given I believe they do have a preference for a small number of highly abundant flowering plants, as flagged in this study.

>> Thank you for this observation. We agree that honeybees commonly forage on a limited set of highly abundant flowering plants. However, we would like to clarify that this behavior reflects their generalist foraging strategy rather than a true preference in the ecological sense. In ecological terms, a "preference" typically implies selection for a resource that is not the most abundant, i.e., choosing disproportionately compared to availability. Honeybees, by contrast, tend to track and exploit the most abundant floral resources, which makes them effective indicators of overall floral availability in the landscape. That said, we agree with your suggestion to reframe the notion of honeybees as a "model" in this context, and we have revised the relevant section of the manuscript accordingly to better reflect this nuance.

There is some lack of clarity and minor suggestions for changes below.

Line 23+: Abstract is missing elements that would be useful, re the dataset, timing, to make it clear that this was one year. This is modelling based on scenarios.

>> We added a sentence in the abstract that summarises the sampling. We hope it addresses your comment adequately. Please see lines 26-28.

Line 38: Some odd phrasing, e.g. how the word ecology is used in the first sentence, I also question putting value on different ecological interactions.

>> We fully agree with this comment and thank you for pointing it out. We have revised the sentence to address your concern and hope it is now clearer. Please see lines 38-39.

Line 66: requires a citation.

>> We added Reference 2 and 19 that support the statement.

Honeybees are generalists but by focusing on this group you miss the deeper flowers that long tongued groups depend on, and the species that have few inflorescences which I think needs mentioning. Managed honeybees collect pollen which can be very easily located and collected making this a useful group to use to investigate. What's more, your findings further suggest that honeybees feed on a small number of presumably abundant flowering plants. This to me might make the unknown element, the impact of the predicted changes in your analysis on flower visiting insect populations alarming as you mention in the discussion. But flags the importance of monitoring impacts on other species, in addition to an understanding of what is available at these locations at the time.

>> We totally agree that the potential wild pollinator impact is alarming and important to investigate. We expanded and highlighted this now more in the discussion.

Line 85: it isn't clear to me whether latitude and temp and/or rain are correlated which is somewhat critical for the interpretation?

>> We thank the reviewer for this insightful observation. In our dataset, temperature and latitude are indeed correlated, as well as precipitation. We agree that this covariation is important when interpreting spatial trends in pollen diversity and climate sensitivity. In our view, this also reinforces the ecological motivation of the study: plant communities, and therefore the pollen collected by honey bees, are already naturally structured by climatic gradients such as temperature and precipitation, which are themselves partly organised along latitude. This inherent covariation is not a confounder to be eliminated, but a fundamental ecological axis of variation that underlies foraging patterns across Europe. We leverage this variation with our sampling design to estimate climate niches.

To reduce misattribution of effects, we included latitude and month of sampling in our linear models rather than temperature and precipitation, which are confounded with geography. In contrast, in the dissimilarity models we partitioned the individual contributions of geography, temperature, and precipitation.

Line 74: 2500 in total, simultaneously every 2 weeks from 310 apiaries, one apiary per location? May-August in 2023 is that then 8 sampling dates? Which is 2480 samples. This could just clarified, and could be clearer in the intro and I think should be stated in the abstract at least the timing as this is important.

>> The number of sampling rounds during the four-month period was nine, not eight. Therefore, the expected number of samples would be 2,790 (instead of 2,480). Of these, we did not obtain data for 290 samples, either due to PCR amplification failure or samples could not be collected for logistic reasons. We have added these details to both the Introduction (lines 75-78) and the Methods section (lines 283-285), as requested.

Line 206: I'm not certain these results (species richness 5-10) indicates a diverse range of collected pollen, some summary stats would be useful to clarify this and to clarify the proportions of the top flowering plant groups that honeybees are feeding on in the results.

>> We thank the reviewer for this observation. We would like to clarify that the values refer to *Effective Species Richness* (e^H), also known as Hill number 1, which incorporates both species richness and evenness of the pollen taxa, as noted in the figure legend. To aid interpretation, we have added a supplementary figure to the GitHub repository displaying the untransformed species richness (i.e., species counts without evenness weighting). Further, a conservative 1% relative abundance threshold was applied at the sample level, based on positive control experiments, to exclude spurious low-frequency taxa. This might also reduce the richness, but we consider this important to reduce the likelihood of introducing contamination into the analyses. We hope this clarifies the range and diversity of the collected pollen.

Line 428: why not a specific date?

>> We added the specific dates (please see lines 283-285), as requested. Thank you for the suggestion.

Line 484: R studio is a platform for analysing data using R software, I think this needs to be referred to appropriately.

>> We changed the sentence to accommodate your comment; Please see lines 348.

Line 482: this indicates pseudo-replication, it's unclear whether this was accounted for in modelling?

>> We thank the reviewer for raising this important concern. Indeed, climate data were only available at monthly resolution per site, so all samples within a given site and month necessarily share the same climate values. While this could introduce pseudo-replication, it is important to note that our niche estimation is based on weighted abundance data and is not implemented as a statistical model with replicated error structure; thus, correction for pseudo-replication is not applicable at that stage. In the final linear models, data were re-aggregated to a single value per site and month, i.e. back to the level at which true original data exists.

Reviewer #4 (Remarks to the Author):

The manuscript under review, "Honey bee food resources under threat from climate change," provides valuable insights into the restoration of plant–animal interactions amidst rapid environmental changes and increasing human interference. It effectively conveys the urgency of monitoring and maintaining the balance of species interactions. The manuscript is generally well-written, with a clear and coherent presentation. However, I have a few concerns and suggestions that I believe could strengthen the study:

Sampling Period: The sampling timeframe appears narrow for drawing conclusions about climate impacts. I recommend extending the study over at least 1–2 additional years to capture interannual variability.

>> We agree that interannual variability is important when investigating long-term ecological trends. However, the inclusion of over 2,500 samples from 27 countries, collected simultaneously over several months, provides robust spatial and seasonal

replication that compensates for the lack of multi-year data. This design enables us to assess plant climatic responses and climate-driven risk patterns across a broad range of real-world conditions — which is the primary aim of the study. While extending the study across multiple years would certainly strengthen assessments of interannual consistency and climatic stability, this was beyond the scope of the current project. The study was funded for one season by the European Commission (DG Environment) and relied heavily on the motivation and coordination of a large network of citizen scientists. Given the unprecedented scale, geographic breadth, and standardization of the sampling effort, we believe the data are well-suited to the study's objectives and provide meaningful, generalizable insights into climate-related risks to floral resources.

Methodological Comments and Suggestions:

Line 92: It would be beneficial to include a main figure showing plant abundance data stratified by location.

>> We appreciate the suggestion. While we agree that spatially stratified abundance data are informative, the scale of the dataset, comprising over 310 locations and up to nine sampling rounds per site, would render such a figure overly dense and difficult to interpret within the scope of the main manuscript. Moreover, such a figure would not directly serve the core research questions addressed here, i.e. climate change risks. That said, the full raw data per location will be made publicly available upon publication, and relative abundance summaries will be released by the European Commission. We are also preparing a follow-up manuscript focused on spatial patterns in floral resource composition, in which location-specific relative abundance data will be presented in greater detail.

Lines 205–207: Could the authors comment on the occurrence of low-abundance pollen taxa and whether they might result from secondary contamination?

>> We thank the reviewer for raising this point. While secondary contamination cannot be entirely ruled out, we took several precautions to minimize its impact and ensure data reliability. First, the bioinformatic pipeline includes multiple quality-control steps, including the removal of chimeric sequences, fungal taxa, and other artifacts. Second, a conservative 1% relative abundance threshold was applied at the sample level, based on positive control experiments, to exclude spurious low-frequency taxa. Third, for all niche-based analyses, we restricted the dataset to plant species that occurred in at least ten independent samples, as noted in the Methods. Finally, most of the analyses rely on aggregated abundance values across broader risk or spatial categories, which reduces the influence of occasional low-abundance taxa. We acknowledge that some low-abundance records, though above the filtering threshold, may reflect scout foraging behavior or incidental collection of wind-dispersed pollen. However, given the multi-layered filtering and aggregation steps, we do not expect these occurrences to meaningfully affect the results. For clarity and cohesion, we have not expanded this discussion in the main text, but would be happy to elaborate further in the Supplementary Information if the editors deem it appropriate.

It would be valuable to include a comparison across multiple bee species to assess species-specific preferences in pollen foraging.

>> We agree that comparing pollen foraging across multiple bee species could yield valuable insights into species-specific floral preferences. However, the focus of this study was on honeybees, the most widely distributed generalist pollinator, and on assessing the climate-related risks to its floral resources at local scales over an entire continent. Including other bee species, particularly wild bees, would introduce substantial variability in geographic distribution, behaviour, floral specialization, phenology, and habitat requirements. Given the diversity of over 2,000 bee species in Europe, applying a standardized sampling protocol across 27 countries would be logistically and ecologically unfeasible. We also believe that our spatially and seasonally broad approach would not be easily transferable to other species, and thus results regarding climate-induced risks would not be directly comparable. Moreover, the study already represents an unprecedented effort in terms of scale, coordination, and sampling detail, and was constrained by time, funding, and available manpower. That said, we fully agree that species-specific comparisons represent a valuable direction for future research, particularly in more localized or targeted studies that can accommodate the ecological diversity of wild bees. We added more details pointed towards such as future perspectives in the discussion.

Line 438: Please clarify how pollen samples were transferred into 2 ml tubes. Also, include a reference for the zirconia bead size-sorting procedure.

>> Thank you for pointing this out. We revised the relevant sentences to incorporate your suggestions. Please see lines 295-297 for details on how pollen samples were transferred into 2 mL tubes, and lines 298-299 for the zirconia bead size-sorting procedure. Although this information was originally described in the *Guideline for Apicultural Citizen Science to Apply the Honey Bee Colony for Bio-monitoring of the Environment* (van der Steen, Pinto, & Quaresma et al.), we have included it here to facilitate reproducibility for readers interested in following the protocol.

Line 445: Consider placing the Maxwell® 440 RSC PureFood kit information here, and include product details in parentheses.

>> Done (please see lines 301-304).

Line 450: The strategy of maintaining equal DNA concentrations for each sample at the start of PCR is commendable. Did this approach apply to all samples, including those with low concentrations?

>> We previously used to diluted the DNA concentration to 10ng/uL, before the 1st PCR stage, when working with smaller sample sets. Inadvertently, a reference to this procedure was left in the Methods section when copying from earlier protocols—our apologies for the oversight. In the present study, due to the large number of samples, we were unable to homogenize DNA concentrations. However, prior tests indicated that this lack of DNA concentration dilution does not impact the results. Moreover, the paper “*Standard*

Methods for Pollen Research” (<https://doi.org/10.1080/00218839.2021.1948240>), specifically Section 8.1, does not recommend DNA dilution as a required step. We removed that piece of information from the Methods section.

Line 454: Please elaborate on how subsamples were selected for indexing. Was pooling involved, and if so, what was the basis? The “1:1 dilution of the first amplicons” requires further clarification.

>> We have revised the sentence to improve clarity; Please see lines 312-314.

Pooling Strategy: How were pools equimolarly combined? Please provide methodological detail.

>> We added “The concentration of each pool was assessed using the Qubit Fluorometric Quantification method (ThermoFisher Scientific)”. Please see line 320-321.

Line 473: The use of 97% sequence similarity—do the authors believe this threshold is sufficient for species-level resolution?

>> We apologize for the lack of clarity — this should have read “at least 97% sequence similarity (top hit).” The final assignment was always based on the top hit, which in most cases was ≥ 99 –100% identical. The sentence has been rephrased for accuracy. The 97% threshold is a commonly applied and well-supported cutoff for ITS2-based plant metabarcoding, particularly in bee-collected pollen studies. While we acknowledge that ITS2 resolution varies among plant taxa, this threshold generally achieves good species-level discrimination for angiosperms and follows established guidelines (e.g., Keller et al. 2021; Sickel et al. 2015; Leonhardt et al. 2022). See below however the reply to the next comment that we also found taxa where this was not valid and we therefore manually reduced the taxonomic assignment level.

Lines 475–477: The method used to merge undetermined taxa with Brassica is unclear. Please elaborate.

>> We apologize for the confusion caused by unclear wording. We did not merge undetermined taxa with *Brassica*. Rather, during manual verification and plausibility checks, we found that several closely related *Brassica* species (e.g., *B. napus*, *B. rapa*, *B. oleracea*, and hybrids) could not be reliably distinguished based on ITS2 sequences due to insufficient divergence in the reference database. Therefore, we chose to reduce the taxonomic resolution for these and treat them collectively as a *Brassica* species complex, to avoid false precision in species-level assignments. We have rephrased the manuscript text accordingly for clarity (Line 338-341).

Line 472: Which database(s) were used for taxonomic assignment?

>> The reference databases used herein were constructed in our previous publication (“Semi-automated sequence curation for reliable reference datasets in ITS2 vascular plant DNA (meta-) barcoding”, by Quaresma et al. 2024 (<https://www.nature.com/articles/s41597-024-02962-5>) and were mentioned in line 475 (line 337-338 in the revised version).

Overall, this manuscript addresses an important and timely topic. With these clarifications and additions, it could make a strong contribution to our understanding of pollinator–plant dynamics under climate stress.

>> Thank you for your valuable comments, which helped us improve the manuscript and enhance its clarity.

Response to Reviewers

Reviewer #1 (Remarks to the Author):

I agree with the authors that various of my concerns have been addressed by careful clarification of the authors' methodologies.

> We appreciate your acknowledgement.

The context I attempted to convey in my review, which the authors found misplaced and counter constructive, seems largely be lost on the authors, which is disappointing. I would have liked to see a finer appreciation for the nuances of applied statistical methodologies and statistical best practices reflected in the authors their work, as it would have improved the quality of the manuscript considerably in my perspective. However, I do not think it crucial for final publication in nature communications.

> Thank you again for this final comment. We would like to reiterate that we believe our applied methods are well suited to the structure and nature of the data and the study's research questions. We nevertheless appreciate the reviewer's perspective and consider these comments valuable impulses for future work. We also thank the reviewer for acknowledgement of the manuscript's readiness for publication.

First and foremost, the authors have provided carefully drafted replies to all my comments, which I appreciate. Some of my concerns the authors shared, some of which they did not, which is their prerogative and requires no further addressing. My main concern, that various analyses were not clearly connected to goals or aims of the study, has been carefully addressed, and the manuscript carefully revised, which has greatly improved the readability and transparency of the study.

> Thank you very much for this positive assessment.

Reviewer #1 (Remarks on code availability):

The authors have chosen to keep their pipeline in a sequential fashion, which I believe to be a poor choice. It does not facilitate reproduction or review, albeit the authors believe their study to be in line with open science initiatives. Be that as it may, in line with open science initiative does not correspond to following "best practices", but as the authors have conveyed unwillingness to change this aspect of their study, there is nothing left to improve in the code.

> It might have gone unnoticed, but we reworked the code in the previous revision so that only a few components remained sequential. Intermediate files are now also stored to facilitate reproducibility, and overall code readability has been much improved. Only sections that are inherently dependent remain sequential, reflecting the sequential character of the programming language used, and we clearly defined dependencies. We are uncertain why this was evaluated as not being in line with "best practices," since the revised code adheres to common standards for transparency, documentation, and reproducibility.

Reviewer #2 (Remarks to the Author):

The Authors did a good job improving their manuscript. All points raised were corrected or clarified. The manuscript now reads much better and the results are clear.

> Thank you very much for your approval and kind evaluation.

Reviewer #3 (Remarks to the Author):

It looks to me like the authors have done an excellent, thorough job of addressing the comments and amending the manuscript.

> Thank you very much for your positive feedback and approval.

Reviewer #4 (Remarks to the Author):

Dear authors,

Thank you for the current revision. The manuscript is suitable for publication now. I have no additional comments.

> Thank you very much for your approval

Reviewer #4 (Remarks on code availability):

Dear authors,

Thank you for the current revision. The manuscript is suitable for publication now. I have no additional comments.

> Thank you very much for your approval